# How to use `frailtypack` for validating failure-time surrogate endpoints using individual patient data from meta-analyses of randomized controlled trials

Casimir Ledoux Sofeu[1,2]*, Virginie Rondeau[1,2]

**1** Biostatistics team, INSERM BPH-U1219, Bordeaux, France, **2** ISPED, Université de Bordeaux, Bordeaux, France

* casimir.sofeu@u-bordeaux.fr, scl.ledoux@gmail.com

## Abstract

### Background and Objective

The use of valid surrogate endpoints can accelerate the development of phase III trials. Numerous validation methods have been proposed with the most popular used in a context of meta-analyses, based on a two-step analysis strategy. For two failure time endpoints, two association measures are usually considered, Kendall's $\tau$ at individual level and adjusted R2 ($\mathrm{adjR}^2_{trial}$) at trial level. However, $\mathrm{adjR}^2_{trial}$ is not always available mainly due to model estimation constraints. More recently, we proposed a one-step validation method based on a joint frailty model, with the aim of reducing estimation issues and estimation bias on the surrogacy evaluation criteria. The model was quite robust with satisfactory results obtained in simulation studies. This study seeks to popularize this new surrogate endpoints validation approach by making the method available in a user-friendly R package.

### Methods

We provide numerous tools in the `frailtypack` R package, including more flexible functions, for the validation of candidate surrogate endpoints using data from multiple randomized clinical trials.

### Results

We implemented the surrogate threshold effect which is used in combination with $R^2_{trial}$ to make decisions concerning the validity of the surrogate endpoints. It is also possible thanks to `frailtypack` to predict the treatment effect on the true endpoint in a new trial using the treatment effect observed on the surrogate endpoint. The leave-one-out cross-validation is available for assessing the accuracy of the prediction using the joint surrogate model. Other tools include data generation, simulation study and graphic representations. We illustrate the use of the new functions with both real data and simulated data.

**Data Availability Statement:** All data files are available from the frailtypack package, which can be download on the Comprehensive R Archive

Network (CRAN). URL: https://cran.r-project.org/web/packages/frailtypack/index.html.

**Funding:** This work was supported by the Association pour la Recherche sur le Cancer, Grant/Award Number: PJA20161205147; Institut National du Cancer, Grant/Award Number: 2017-125; and Institut national de la santé et de la recherche médicale; Région Aquitaine. The funders had no role in study design, data collection and analysis, decision to publish, or preparation of the manuscript.

**Competing interests:** The authors have declared that no competing interests exist.

## Conclusion

This article proposes new attractive and well developed tools for validating failure time surrogate endpoints.

## Introduction

The choice of endpoint for assessing the efficacy of a new treatment is a key step in setting up clinical trials. The use of the true endpoint increases the cost and duration of trials, and usually induces an alteration of the treatment effects over time [1, 2]. For example, in oncology, overall survival is a common clinical endpoint used during phase 3 trials to evaluate the clinical benefit of new treatments. However, its use requires a sufficiently long follow-up time and a sufficiently high sample size to show a significant difference in the treatment effect. To overcome this problem, there has been a lot of interest over the last three decades in the use of alternative criteria or surrogate endpoints to reduce the cost and shorten the duration of phase 3 trials [1–4]. A good surrogate endpoint should predict the effect of treatment on the primary endpoint [3].

Prentice (1989) [5] enumerated four criteria to be fulfilled by a putative surrogate endpoint. The fourth criterion, often called Prentice's criterion, stipulates that a surrogate endpoint must capture the full treatment effect upon the true endpoint. The validation of Prentice's criterion based on a clinical trial was quite difficult, mainly due to a lack of power and the difficulty to verify an assumption related to the relation between the treatment effects upon the true and the surrogate endpoints. Therefore, to verify this assumption and obtain a consistent sample size, Buyse *et al.* (2000) [6] like other authors [7] suggested basing validation on the meta-analytic (or multicenter) data. An important point when dealing with meta-analytic data is to take heterogeneity between trials into account, for the purpose of prediction outside the scope of the trial. Thus, a validated surrogate endpoint from meta-analytic data can be used to predict the treatment effect upon the true endpoint in any trial.

In the meta-analysis framework, when both the surrogate and the true endpoints are failure times, the current consensus is to base validation on the two-stage analysis strategy proposed by Burzykowski *et al.* [8]. In the first stage, the association between the surrogate and true endpoints is evaluated using a bivariate copula model after taken the trial specific treatment effects into account. In the second stage, the prediction of the treatment effect on the true endpoint based on the observed treatment effect on the surrogate endpoint is assessed using the adjusted coefficient of determination (adj$R^2_{trial}$). adj$R^2_{trial}$ is obtained from the regression model on the estimates of the trial-specific treatment effects on both the surrogate and the true endpoints, after adjusting on the estimation errors obtained in the first-stage model. The programs that implement this method are available in the R package `surrosurv` [9] and the SAS macro `%COPULA` [10]. However, the practical use of the two-stage copula model is often difficult, mainly due to convergence issues or model estimation with the adjustment on the estimation errors [11–13]. This drawback led to the development since Burzykowski *et al.* [8] of alternative approaches [11, 13–17].

Most of the novel methods, except that of Sofeu *et al.* [17] and Rotolo *et al.* [13], are based on a two-stage validation strategy. Alonso and Molenberghs [14] proposed an information theory approach, with a new definition and quantification of surrogacy at the individual level and the trial level. The drawback of this method was the difficulty to provide a hard cut-off value in

the information-theoretic measure, to discriminate between good and bad surrogates. Buyse et al. [15] suggested a two-stage validation approach in which individual-level surrogacy was evaluated through the association between the trial-specific Kaplan-Meier estimates of the true endpoint versus Kaplan-Meier estimates of the surrogate endpoint at a fixed time point. It is also possible to base validation at the individual level on a bivariate copula model. In the trial-level evaluation, a weighted linear regression on the treatment effects on the surrogate and true endpoints was fitted and the coefficient of determination ($R^2$) was used to quantify the proportion of variance explained by the regressions. The available programs also make it possible to account for variability between trials using a robust sandwich estimator of Lin and Wei [18].

For the approaches described in the previous paragraph, the R package `surrogate` [19], the SAS macros `%TWOSTAGECOX` and `%TWOSTAGEKM`, and the SAS programs available in Alonso et al. [10] were provided to carry out the evaluation exercise. Rotolo et al. [13] proposed a one-step validation approach based on auxiliary mixed Poisson models, which employs a bivariate survival model with an individual random effect shared between the two endpoints and correlated treatment-by-trial interactions. Simulation results described by the authors showed estimation biases on the surrogacy assessment measures, especially in the event of a high association and when heterogeneity of baseline risk is taken into account. The associated program was implemented in the R package `surrosurv` [9]. Renfro et al. [11] suggested estimating the second-stage model in a Bayesian framework and the estimate of the adjusted $R^2_{trial}$ was then based on the posterior distribution of the parameters of the adjusted model. The corresponding trial-level surrogacy can be evaluated by adapting the `WinBUGS` and `R` programs described in Bujkiewicz et al. [20]. This approach emphasizes a decrease in estimation performance of the adjusted $R^2_{trial}$ when the data characteristics are close to reality (for example, low trial size or number of trial).

More recently, we proposed a one-step validation approach based on a joint frailty model [17] to reduce convergence issues and estimation biases on the surrogacy evaluation criteria. In this novel method, we used a flexible form of the baseline hazard functions using splines to obtain smooth risk functions, which represent incidence in epidemiology. Several integration strategies were considered to compute integrals over the random effects, present in the marginal log-likelihood. The proposed joint surrogate model showed satisfactory results compared to the existing two-step copula and one-step Poisson approaches.

We aim in this paper to popularize this new surrogate endpoints validation approach by making the method available in a user-friendly R package (`frailtypack`). We have developed a prediction tool for the treatment effect on true endpoints based on the observed treatment effect on surrogate endpoints. Interpretation of $R^2_{trial}$ and decision-making about the validity of the candidate surrogate endpoint are possible thanks to the classification suggested by the Institute for Quality and Efficiency in Health Care (IQWiG) [21], and surrogate threshold effect (STE) introduced by Burzykowski and Buyse [22]. Other tools are for displaying the basic risks and survival functions, for model assessment, and for data generation based on the joint surrogate model. Another attractive goal of this article is to provide a tool to perform simulation studies.

`frailtypack` is an R package that fits a variety of frailty models containing one or more random effects, or shared frailty. It includes a shared frailty model, a joint frailty model for recurrent events and terminal event, others forms of advanced joint frailty models [23], and now a joint frailty model for evaluating surrogate endpoints in meta-analyses of randomized controlled trials with failure-time endpoints. In this paper we focus on a particular subset of features applicable for evaluating surrogate endpoints.

The rest of this paper is organized as follows. In the next section, we summarize the joint surrogate model with the estimation methods and the surrogacy evaluation criteria. We end it with the definition of STE. In the third section, we introduce the functions developed in the R-package `frailtypack` to estimate the parameters of the joint surrogate model, as well as the new functions related to the surrogacy evaluation. In the fourth section, we illustrate the new functions using generated data and individual patient data from the Ovarian Cancer Meta-Analysis Project [24]. Finally, we present a concluding discussion.

## Methodology

In this section, we present the one-step joint surrogate model for evaluating a candidate surrogate endpoint [17]. The model estimation and the surrogacy evaluation criteria are also discussed here.

### Model and estimation

**Joint surrogate model definition.** Let us consider data from a meta-analysis (or a multi-center study); let $S_{ij}$ and $T_{ij}$ be two time-to-event endpoints associated respectively with the surrogate endpoint and the true endpoint such that $S_{ij} < T_{ij}$ or $S_{ij} = T_{ij}$ in the event of right censoring. We denote $Z_{ij1}$ the treatment indicator. $S_{ij}$ can be the progression-free survival time (defined as the time from randomization to clinical progression of the disease or death) in patients treated for cancer and $T_{ij}$ the overall survival (defined as the time from randomization to death from any cause). For the $j^{th}$ subject ($j = 1, \ldots, n_i$) of the $i^{th}$ trial ($i = 1, \ldots, G$), the joint surrogate model is defined as follows [17]:

$$\begin{cases} \lambda_{S,ij}(t|\omega_{ij}, u_i, v_{S_i}, Z_{ij1}) &= \lambda_{0S}(t)\exp(\omega_{ij} + u_i + v_{S_i}Z_{ij1} + \beta_S Z_{ij1}) \\ \lambda_{T,ij}(t|\omega_{ij}, u_i, v_{T_i}, Z_{ij1}) &= \lambda_{0T}(t)\exp(\zeta\omega_{ij} + \alpha u_i + v_{T_i}Z_{ij1} + \beta_T Z_{ij1}) \end{cases} \tag{1}$$

where,

$$\omega_{ij} \sim N(0, \theta), u_i \sim N(0, \gamma), \omega_{ij} \perp u_i, u_i \perp v_{S_i}, u_i \perp v_{T_i}$$

and

$$\begin{pmatrix} v_{S_i} \\ v_{T_i} \end{pmatrix} \sim MVN(\mathbf{0}, \Sigma_v), with \ \Sigma_v = \begin{pmatrix} \sigma^2_{v_S} & \sigma_{v_{ST}} \\ \sigma_{v_{ST}} & \sigma^2_{v_T} \end{pmatrix}$$

In this model, $\lambda_{0S}(t)$ is the baseline hazard function associated with the surrogate endpoint and $\beta_S$ the fixed treatment effect (or log-hazard ratio); $\lambda_{0T}(t)$ is the baseline hazard function associated with the true endpoint and $\beta_T$ the fixed treatment effect. $\omega_{ij}$ is a shared individual-level frailty that serve to take into account the heterogeneity in the data at the individual level due to unobserved covariates; $u_i$ is a shared frailty effect associated with the baseline hazard function that serve to take into account the heterogeneity between trials of the baseline hazard function, associated with the fact that we have several trials in this meta-analytical design. Coefficients $\zeta$ and $\alpha$ distinguish both individual and trial-level heterogeneities between the surrogate and the true endpoint. $v_{S_i}$ and $v_{T_i}$ are two correlated random effects treatment-by-trial interactions.

**Estimation.** **Marginal log-likelihood** Let $\delta_{ij}$ and $\delta^*_{ij}$ be the progression and the death indicators. Sofeu *et al.* [17] showed that the marginal log-likelihood from model (1) includes two

integration levels and is defined as follows:

$$l(\Phi) = \log\left\{\prod_{i=1}^{G}\int_{U}\left[\prod_{j=1}^{ni}\int_{\omega_{ij}}\lambda_{Sij}^{\delta_{ij}}\cdot S(S_{ij})\cdot\lambda_{Tij}^{\delta_{ij}^{*}}\cdot S(T_{ij})f(\omega_{ij})d\omega_{ij}\right]f(v_{S_i},v_{T_i})f(u_i)dU\right\} \quad (2)$$

where $\Phi = (\hat{\sigma}_{v_S}^2, \hat{\sigma}_{v_T}^2, \hat{\sigma}_{v_{ST}}, \hat{\theta}, \hat{\gamma}, \hat{\lambda}_{0T}(.), \hat{\lambda}_{0S}(.), \hat{\beta}_S, \hat{\beta}_T)$ is the vector of the model parameters and $U = (u_i, v_{S_i}, v_{T_i})$ is the vector of trial random effects. $\hat{\lambda}_{0S}(.)$ and $\hat{\lambda}_{0T}(.)$ are estimates for the baseline hazard functions associated with the surrogate endpoint and the true endpoint.

**Parameters estimation** The model parameters $\Phi$ were estimated by a semi-parametric approach using the maximization of the penalized likelihood. We used the robust Marquardt algorithm [25], which is a mixture between the newton-Raphson and the steepest descent algorithm. For more details on the penalized likelihood, see the S1A Appendix in S1 Appendix or [26]. In order to estimate the integrals present in (2), different numerical integration strategies were considered, including a mixture of the Monte-Carlo integration with the Pseudo-adaptive or the classical Gauss-Hermite quadrature.

## Surrogacy evaluation criteria and interpretation

We have already proposed new definitions of Kendall's $\tau$ and coefficient of determination as individual-level and trial-level association measures to evaluate a candidate surrogate endpoint [17]. We recall in the S1B and S1C Appendix in S1 Appendix the formulation of these association measures.

## Prediction and surrogate threshold effect (STE)

Gail *et al.* [27] underlined some issues in using $R_{trial}^2$ for assessing a candidate surrogate endpoint. The first problem is the difficulty in interpreting $R_{trial}^2$. For perfect prediction of the treatment effect on the true endpoints, $R_{trial}^2$ must be equal to 1. However, such a situation is impossible in practice. Therefore, for $R_{trial}^2 \neq 1$, it is not clear what threshold would be sufficient for a valid surrogate endpoint. Another problem raised by Gail *et al.* [27] is that, unless $R_{trial}^2 = 1$, the variance of the prediction of the treatment effect on the true endpoint in a new trial cannot be reduced to 0, even in the absence of any estimation error in the trial. Furthermore, if this effect is estimated directly from data on the true endpoint, this estimation error can theoretically be made arbitrarily close to 0 by increasing the trial's sample size. To address these issues, Burzykowski and Buyse [22] proposed a new concept, the surrogate threshold effect. One of the most interesting features of STE is its natural interpretation from a clinical point of view. STE represents the minimum treatment effect on the surrogate necessary to predict a non-zero (significant) effect on the true endpoint. We show in S1D Appendix in S1 Appendix that STE, based on model (1), can be obtained by solving one of the following quadratic equations:

$$E(\beta_T + v_{T0}|\beta_{S_0}, \vartheta) - z_{1-(\gamma/2)}\sqrt{Var(\beta_T + v_{T0}|\beta_{S_0}, \vartheta)} = 0 \quad (3)$$

for the lower prediction limit function of the treatment effect on the true endpoint based on the observed treatment effect on the surrogate endpoint, or

$$E(\beta_T + v_{T0}|\beta_{S_0}, \vartheta) + z_{1-(\gamma/2)}\sqrt{Var(\beta_T + v_{T0}|\beta_{S_0}, \vartheta)} = 0, \quad (4)$$

for the upper prediction limit function. Elements in Eqs (3) and (4) are defined in S1D Appendix in S1 Appendix.

Readers can refer to S1E Appendix in S1 Appendix for the interpretation of STE, in combination with $R^2_{trial}$ and decision-making as suggested by the German Institute for Quality and Efficiency in Health Care [21]

## Available functions in the `frailtypack` R package for surrogacy evaluation

In this section, we introduce the new R functions, used to estimate model (1). Functions for data generation and simulation studies are also described.

### Estimation of joint surrogate model and surrogacy evaluation

**The `jointSurroPenal()` function.** Model (1) can be fitted using the `jointSurroPenal()` function defined as follows:

```
jointSurroPenal(data, maxit = 40, indicator.zeta = 1, indica-
tor.alpha = 1,
    frail.base = 1, n.knots = 6, LIMlogl = 0.001, LIMparam = 0.001,
    LIMderiv = 0.001, nb.mc = 300, nb.gh = 32, nb.gh2 = 20,
adaptatif = 0,
    int.method = 2, nb.iterPGH = 5, nb.MC.kendall = 10000,
    nboot.kendall = 1000, true.init.val = 0, theta.init = 1,
  sigma.ss.init = 0.5, scale = 1, sigma.tt.init = 0.5, sigma.st.
init = 0.48,
    gamma.init = 0.5, alpha.init = 1, zeta.init = 1, betas.
init = 0.5,
    betat.init = 0.5, random.generator = 1, kappa.use = 4,
random = 0,
    seed = 0, random.nb.sim = 0, init.kappa = NULL, nb.decimal = 4,
    print.times = TRUE, print.iter = FALSE)
```

The mandatory argument of this function is `data`, the dataset to use for the estimations. Argument `data` refers to a dataframe including at least 7 variables: `patientID`, `trialID`, `timeS`, `statusS`, `timeT`, `status` and `trt`. The description of these variables, like other arguments of the function, can be found in S2A Appendix in S2 Appendix, or via the R command help(jointSurroPenal). The rest of the arguments can be set to their default values. In addition, details on the required arguments/values are given in the illustration section.

**The `jointSurroPenal` object.** The function `jointSurroPenal()` returns an object of class '`jointSurroPenal`', if the joint surrogate model has been estimated. We describe in S2A Appendix in S2 Appendix some of the relevant returned values, as well as the functions which can be applied to this object. A full description can be found by displaying the help on the function `jointSurroPenal()`.

### Data generation using the R function `jointSurrSimul()`

For data generation purposes, we implemented the algorithm described in Sofeu *et al.* [17] in the R function `jointSurrSimul()`. The generation procedure is based on model (1). A variant of this algorithm is to base generation on a model that includes just a shared frailty term at the individual level as described by Rondeau *et al.* [28]. This function is defined as follows:

```
jointSurrSimul(n.obs = 600, n.trial = 30, cens.adm = 549.24,
alpha = 1.5,
```

```
        theta = 3.5, gamma = 2.5, zeta = 1, sigma.s = 0.7, sigma.
t = 0.7,
        rsqrt = 0.8, betas = -1.25, betat = -1.25, frailt.base = 1,
        lambda.S = 1.8, nu.S = 0.0045, lambda.T = 3, nu.T = 0.0025,
ver = 1,
        typeOf = 1, equi.subj.trial = 1, equi.subj.trt = 1,
        prop.subj.trial = NULL, full.data = 0, prop.subj.trt = NULL,
        random.generator = 1, random = 0, random.nb.sim = 0, seed = 0,
        nb.reject.data = 0)
```

Arguments of the `jointSurrSimul()` function are accessible using the R command `help(jointSurrSimul)`. An exhaustive description is presented in S2B Appendix in S2 Appendix.

## Simulation studies based on the joint surrogate model

It is possible to perform simulation studies based on model (1), using the function `jointSurroPenalSimul()` defines as follows:

```
    jointSurroPenalSimul(nb.dataset = 1, nbSubSimul = 1000,
ntrialSimul = 30,
        equi.subj.trial = 1, prop.subj.trial = NULL, equi.subj.trt = 1,
        prop.subj.trt = NULL, theta2 = 3.5, zeta = 1, gamma.ui = 2.5,
        alpha.ui = 1, sigma.s = 0.7, sigma.t = 0.7, R2 = 0.81, betas =
-1.25,
        betat = -1.25, lambdas = 1.8, nus = 0.0045, lambdat = 3,
nut = 0.0025,
        time.cens = 549, indicator.zeta = 1, indicator.alpha = 1,
frail.base = 1,
        init.kappa = NULL, n.knots = 6, maxit = 40, LIMparam = 0.001,
        LIMlogl = 0.001, LIMderiv = 0.001, int.method = 2,
adaptatif = 0,
        nb.iterPGH = 5, nb.mc = 300, nb.gh = 32, nb.gh2 = 20,
        nb.MC.kendall = 10000, nboot.kendall = 1000, true.init.val = 0,
        theta.init = 1, zeta.init = 1, gamma.init = 0.5, alpha.
init = 1,
        sigma.ss.init = 0.5, sigma.tt.init = 0.5, sigma.st.init = 0.48,
        betas.init = 0.5, betat.init = 0.5, kappa.use = 4,
        random.generator = 1, random = 0, random.nb.sim = 0, seed = 0,
        nb.decimal = 4, print.times = TRUE, print.iter = FALSE)
```

Most of the arguments in this function are mandatory for the user, taking into account the simulation design. S2B Appendix in S2 Appendix describes all the arguments, as well as the elements of the 'jointSurroPenalSimul' object.

## Kendall's $\tau$ estimation using the function **jointSurroTKendall**

The function `jointSurroTKendall()` is used to estimate Kendall's $\tau$ described in S1B Appendix in S1 Appendix, based on the estimates from the model (1). It is possible to perform the numerical integration with the Monte-Carlo or the Gauss-Hermite quadrature method. The `jointSurroTKendall()` function is defined as shown below, with arguments described in S2D Appendix in S2 Appendix. This function returns the estimated value of Kendall's $\tau$

```
jointSurroTKendall(object = NULL, theta, gamma, alpha = 1,
zeta = 1,
    int.method = 0, sigma.v = matrix(rep(0, 4), 2, 2), nb.gh = 32,
    nb.MC.kendall = 10000, random.generator = 1, random.nb.sim = 0,
    random = 0, seed = 0, ui = 1)
```

## Illustrations

### Computational details and package installation

Estimations in the proposed functions are based on `Fortran` programs, with parallel computing using `OpenMP`, to speed up calculations. Thus, we used R as an interface between the user and the `Fortran` compiler. The stable version of `frailtypack` is available on the Comprehensive R Archive Network (CRAN) [29]. Furthermore, the ongoing version can be found on GitHub at https://github.com/socale/frailtypack. A list of other models implemented in `frailtypack` [23] can be found in S1 Fig. The results in this paper were obtained using R version `3.5.2` and the `frailtypack` package version 3.0.3, using a processor `Intel (R) Xeon(R) CPU E5-2690 v2 @ 3.00GHz` including 40 cores and a Read Only Memory (RAM) of 378 Gb. A standard laptop and a desktop PC under recent versions of R can be used to fit the model. The results will be the same, but with longer computing time. For example, using a standard desktop PC in the application, the fit took around 1 hours compared to 9 min with a server including 40 cores and a RAM of 378 Go.

The `frailtypack` package can be installed in any R session using the `install. packages` command as follows:

```
install.packages ("frailtypack", dependencies = T, type =
"source",
repos = "https://cloud.r-project.org")
```

Installation via `GitHub` is possible thanks to the `devtools` package. All dependencies required by `frailtypack` must be installed first. The installation commands are:

```
install.packages (c("survC1", "doBy","statmod"), repos =
"https://cloud.r-project.org")
devtools::install github ("socale/frailtypack", ref = "sur-
rogacy submetted 3-0-3")
```

Finally, `frailtypack` must be loaded using the command:

```
library (frailtypack)
```

### Data source

We illustrate the use of the developed functions with the individual patient data of the Ovarian Cancer Meta-Analysis Project [24] and a generated dataset based on model (1). We also describe the simulation studies at the end of this section.

**Description of `dataOvarian` dataset.** The `dataOvarian` dataset combines data that were collected in four double-blind randomized clinical trials in advanced ovarian cancer. In

the first two trials of this study, data were available on the centers in which patients were treated, and each of the two trials were considered as a homogeneous group according to the investigators. Finally, the statistical unit in the first two trials was center and it was trial for the last two trials. Therefore, a total of 50 units were available for surrogacy evaluation. The objective in these studies was to examine the efficacy of cyclophosphamide plus cisplatin (CP) versus cyclophosphamide plus adriamycin plus cisplatin (CAP) to treat advanced ovarian cancer. The candidate surrogate endpoint **S** was progression-free survival time (PFS), defined as the time (in years) from randomization to clinical progression of the disease or death. The true endpoint **T** was survival time, defined as the time (in years) from randomization to death from any cause. The dataset includes 1192 subjects with 82% of PFS-related events at a median survival time of 78.7 days [Interquartile range (IQR): 36.6–202.5], and 79.8% of deaths at a median survival time of 111.4 days [IQR: 56.0–275.9]. Data can be loaded as follows:

```
data ("dataOvarian", package = "frailtypack")
```

By displaying the structure of this dataset, we can find the same structure as in the function `jointSurroPenal()`, with 7 variables. The column `trialID` here refers to the analysis unit.

```
str (dataOvarian)
  'data.frame': 1192 obs. of 7 variables:
  $ patientID:  int 1 2 3 4 5 6 7 8 9 10 ...
  $ trialID  :  num 2 2 2 2 2 2 2 2 2 2 ...
  $ trt      :  int 0 0 0 1 0 1 0 0 1 1 ...
  $ timeS    :  num 0.1052 0.8952 0.079 1.7393 0.0913 ...
  $ statusS  :  int 1 1 1 0 1 1 1 1 1 1 ...
  $ timeT    :  num 0.186 1.409 0.126 1.739 0.127 ...
  $ statusT  :  int 1 1 1 0 1 1 1 1 1 1 ...
```

**Generated dataset.** In the example below, we generate a meta-analysis including $600$ subjects in $30$ trials. Arguments $\alpha$, $\theta$, $\zeta$ and $\gamma$ are fixed to obtain a Kendall's $\tau$ of $0.61$, which is obtained using the `jointSurroTKendall()` function as follows:

```
jointSurroTKendall (theta = 3.5, gamma = 2.5, alpha = 1.5,
zeta = 1)
   [1] 0.6062975
```

Otherwise, the trial level surrogacy, $R^2_{trial}$ is fixed to $0.8$. This could correspond to simulation design including high trial level and high individual level surrogacy. The treatment effects $\beta_S$ and $\beta_T$ are set to $-1.25$ to consider protective effects both on the surrogate endpoint and the true endpoint. The code below is used to generate the dataset using the `jointSurrSimul()` function introduce d previously, and display the head.

```
data.sim <- jointSurrSimul(n.obs = 600, n.trial = 30,
alpha = 1.5,
  theta = 3.5, gamma = 2.5, zeta = 1, sigma.s = 0.7, sigma.
t = 0.7,
```

```
   rsqrt = 0.8, betas = -1.25, betat = -1.25, random.
generator = 1,

   seed = 0, nb.reject.data = 0)

head (data.sim)

  patientID  trialID trt      timeS  statusS      timeT  statusT
1          1        1   0   8.243721       1   38.41068        1
2          2        1   1 446.169009       0  446.16901        1
3          3        1   1 110.418853       0  110.41885        1
4          4        1   1  70.262075       0   70.26207        1
5          5        1   1 382.973632       1  549.24000        0
6          6        1   0  61.148254       1  230.24486        1
```

## Surrogacy evaluation

In this section, we use the dataset previously described to illustrate the evaluation of the surrogate endpoints based on the one-step joint surrogate model ([1]). Different arguments of the associated functions will be explored as the returned values.

**Model estimation based on the advanced ovarian cancer meta-analysis dataset.** From a practical point of view, the most important arguments for using the `jointSurroPenal()` function beyond the standard argument (`data`) concern the following: the parametrization of the model (with arguments `indicator.zeta` and `indicator.alpha`), the method of integration and associated arguments (`int.method, n.knots, nb.mc, nb.gh, nb.gh2, adaptatif`), the smoothing parameters (`init.kappa` and `kappa.use`) and the scale of survival times (`scale`). Although optional, all these arguments can be used to manage the convergence issues. The choice of the values to assign to these arguments can be based on the convergence of model. When the convergence issues are fixed, users can implement the likelihood cross-validation criteria to evaluate the goodness of fit of different models, as shown later in this section. In the first step, users can try the model with the default values.

In the event of convergence issues, we recommend the following strategy: Changing the number of samples for Monte-Carlo integration (`nb.mc`) by choosing a numerical value between `100` and `300`; varying the number of nodes for the Gaussian-Hermite quadrature integration (`nb.gh` and `nb.gh2`) by choosing the values between `15`, `20` and `32`; varying the number of nodes for spline (`n.knots`) by a numerical value between `6` and `10`; providing new values for the smoothing parameters (`init.kappa`). Users can also set the arguments $\alpha$ or $\zeta$ to `1` (`indicator.zeta = 1` or `indicator.alpha = 1`) to avoid estimating these parameters. We also recommend changing the integration method with the arguments `int.method` and `adaptatif`. For example, by using `adaptatif = 1` for integration over the random effects at the individual level, one could use the pseudo-adaptive quadrature Gaussian-Hermite integration instead of the classical quadrature Gaussian-Hermite method. By changing the scale of the survival times (argument `scale`) and considering years instead of days, it is possible to solve some of the numerical issues.

Using the default values based on the advanced ovarian cancer dataset, the model did not converge. By changing the value of some arguments, we obtained the following set of arguments/values which allowed convergence:

```
joint.surro.ovar <- jointSurroPenal(data = dataOvarian, n.
knots = 8,

    indicator.alpha = 0, nb.mc = 200, scale = 1/365)
```

In this model, we fix the coefficient $\alpha$ to $1$, and thereby do not estimate it. We consider $8$ spline nodes for the baseline hazards. By default, we use the fixed initial values and obtain smoothing parameters by cross-validation on reduced models. We approximate integrals over the random effects using a combination of Monte-Carlo with $200$ samples and classical Gauss-Hermite quadrature with $32$ nodes. To solve numerical problems during estimation, we re-scale the survival times by converting days to years. This parametrization of the model provided the results described in the next section.

**Summary of results.** By applying the function `summary()` on the object `joint.surro.ovar`, the following results are displayed in the event of convergence:

```
summary(joint.surro.ovar)
  Estimates for variance parameters of random effects

          Estimate  Std Error       z        P
  theta       6.848     0.3786  18.086   < e-10  ***
  zeta        1.792     0.0714  25.095   < e-10  ***
  gamma       0.045     0.0774   0.576   0.5645
  sigma2_S    0.610     0.3733   1.633   0.1025
  sigma2_T    1.830     1.0202   1.794   0.07287  .
  sigma_ST    1.056     0.6067   1.741   0.0817   .
  Estimates for fixed treatment effects

          Estimate  Std Error       z        P
  beta_S    -0.596     0.2298  -2.595  0.009463 **
  beta_T    -0.841     0.3936  -2.136   0.03264  *

  ---

  Signif. codes: 0 '***' 0.001 '**' 0.01 '*' 0.05 '.' 0.1 ' ' 1

  hazard ratios (HR) and confidence intervals for fixed treatment effects

          exp(coef)  Inf.95.CI  Sup.95.CI
  beta_S      0.551      0.351      0.864
  beta_T      0.431      0.199      0.933
```

```
Surrogacy evaluation criterion

              Level Estimate Std Error Inf.95.CI Sup.95.CI Strength

Ktau    Individual    0.683        --       0.664     0.696

R2trial      Trial    1.000     0.001       0.998     1.002      High

R2.boot      Trial    0.982        --       0.896     1.000      High

---

Association strength: <= 0.49 'Low';]0.49-0.72['Medium'; >= 0.72 'High'

---

Surrogate threshold effect (STE): -0.273 (HR = 0.761)

Convergence parameters

Penalized marginal log-likelihood = -10892.611

Number of iterations = 29

LCV = the approximate likelihood cross-validation criterion

     in semi-parametrical case = 9.162

Convergence criteria:

  parameters = 9.573e-06 likelihood = 8.426e-08 gradient = 4.507e-08
```

The results are organized in five parts. We first present estimates for the variance parameters of the random effects and the coefficients $\zeta$ and $\alpha$ (if applicable). This includes standard errors, z-statistics and $p$ value of the Wald test. Results suggest a strong heterogeneity at the individual level, observed on the endpoints ($\theta = 6.848$ compared to 0), and more pronounced on the true endpoint ($\zeta = 1.792$ compared to 1). The estimated value of $\gamma$ suggests homogeneous baseline hazards across trials ($\gamma = 0.045$, $p > 0.5$), both on the surrogate endpoint and on the true endpoint. This could explain the identification problem encountered by considering the coefficient $\alpha$ in the model. The parameters $\sigma_S^2$, $\sigma_T^2$, $\sigma_{ST}$ suggest the presence of heterogeneity at trial level interacting with the treatment ($p < = 0.10$). The next two parts of the results show estimates for the fixed treatment effects $\beta_S$ given the random effects ($u_i$, $v_{S_i}$) and $\beta_T$ given ($u_i$, $v_{T_i}$), with the associated hazard ratios and confidence intervals. These parameters can be interpreted as usual, but taking adjustment on the random effects into account. We observed significant protective effects of the treatment on the surrogate endpoint and on the true endpoint ($p < 0.05$).

The fourth part of the results describes the surrogacy evaluation criterion. Kendall's $\tau$, $R_{trial}^2$ and $R_{trial,boot}^2$ (obtained using parametric bootstrap) are available with the associated confidence intervals as is the standard error of $R_{trial}^2$ obtained by the Delta method [30]. Arguments `int.method.kt` and `nb.gh` of the `summary()` function can be used to choose between the Monte-Carlo and the Gauss-Hermite quadrature which integration method is to be used to estimate Kendall's $\tau$, and set the number of quadrature nodes when appropriate. Using at least 500 samples for the Monte-carlo integration and at least 15 quadrature nodes the two integration methods generally yield the same results for Kendall's $\tau$.

These results suggest high association measurement at the individual level (Kendall's $\tau$ = 0.68 [0.66–0.70]), and high correlation strength at the trial level ($R_{trial,boot}^2 = 0.98$ [0.90–1.00])

between the surrogate endpoint and the true endpoints, according to the classification of the surrogacy criteria proposed by the Institute of Quality and Efficiency in Health Care [31, 32]. Given that Kendall's $\tau$ is adjusted on random effects at the individual level [17], it is quite difficult to observe a value > `0.7` compared to unadjusted ones from the two-step copula approach of Burzykowski `et al.` [8]. A very high value suggests extreme values for the parameters $\alpha$, $\zeta$, $\theta$ or $\gamma$, although such values are difficult to observe in practice. Therefore, a value `around 0.65` can be considered as sufficient for validating surrogacy at the individual level.

We also compute and display the surrogate threshold effect with the associated hazard risk. We obtain an acceptable value of STE (- 0.273, HR = 0.761), which illustrates the high validity of the surrogate. As mentioned by [22], unrealistically large/small values of STE (e.g., corresponding to a HR of less than 0.5) would indicate too wide prediction limits and, consequently, poor validity of the surrogate. Therefore, as observed previously [8], PFS can be considered as a valid surrogate endpoint for OS when evaluating new treatments for advanced ovarian cancer.

The last part of the results describes the convergence parameters.

**Model estimation based on generated dataset.** Here, we estimate two joint surrogate models for the purpose of model comparison, based on the generated dataset `data.sim`. Integrals are approximated using a combination of Monte Carlo and classical Gauss-Hermite in the first model and a combination of Monte Carlo and Pseudo-adaptive Gauss-Hermite integration in the second one. The codes for both models are described as follows:

```
joint.surro.sim.MCGH <- jointSurroPenal(data = data.sim,
int.method = 2,

    nb.mc = 300, nb.gh = 20)

joint.surro.sim.MCPGH <- jointSurroPenal(data = data.sim,
int.method = 2,

    nb.mc = 300, nb.gh = 20, adaptatif = 1)
```

A relevant question in this case might be how to compare different models, or how to choose the optimal value of the number of knots for spline, the number of quadrature points, the number of samples for Monte-Carlo, or the optimal integration method. We propose in this package to base comparison on the approximated likelihood cross-validation criterion. The lower the value obtained for this parameter, the better the associated model will be.

**Choice of model based on `LCV`.** The LCV for models `joint.surro.sim.MCGH` and `joint.surro.sim.MCPGH` are respectively

```
joint.surro.sim.MCGH$LCV

  [1] 8.29982

joint.surro.sim.MCPGH$LCV

  [1] 8.31713
```

As expected [17], the two observed values of LCV are quite similar. The `summary()` function applied to previous objects give results shown below. When comparing the two models, estimates of most coefficients and standard errors showed some differences. However this observation does not alter conclusions on the surrogacy validity captured by Kendall's $\tau$ and $R^2_{trial}$.

```
summary(joint.surro.sim.MCGH)
  Estimates for variance parameters of random effects
          Estimate  Std Error       z        P
  theta      3.450    0.4928     7.001    < e-10   ***
  zeta       1.506    0.2364     6.369  1.899e-10  ***
  gamma      1.881    0.5602     3.358  0.0007853  ***
  alpha      0.916    0.1443     6.348  2.183e-10  ***
  sigma2_S   0.703    0.4289     1.640    0.1011
  sigma2_T   1.096    0.6147     1.783    0.07451   .
  sigma_ST   0.442    0.3974     1.113    0.2657
  Estimates for fixed treatment effects
          Estimate  Std Error      z        P
  beta_S   -2.046     0.2667   -7.673    < e-10 ***
  beta_T   -1.844     0.3562   -5.177  2.25e-07 ***
  ---
  Signif. codes: 0 '***' 0.001 '**' 0.01 '*' 0.05 '.' 0.1 ' ' 1
  hazard ratios (HR) and confidence intervals for fixed treatment effects
        exp(coef) Inf.95.CI Sup.95.CI
  beta_S    0.129     0.077     0.218
  beta_T    0.158     0.079     0.318
  Surrogacy evaluation criterion
             Level Estimate Std Error Inf.95.CI Sup.95.CI Strength
  Ktau    Individual   0.596      --      0.542     0.625
  R2trial      Trial   0.254    0.276    -0.288     0.796      Low
  R2.boot      Trial   0.290      --      0.002     0.767      Low
  ---
  Association strength: <= 0.49 'Low';]0.49-0.72['Medium'; >= 0.72 'High'
  ---
  Surrogate threshold effect (STE): -8.523 (HR = 0)
  Convergence parameters
  Penalized marginal log-likelihood = -4957.842
  Number of iterations = 14
  LCV = approximate likelihood cross-validation criterion
       in the semi-parametrical case = 8.3
```

```
Convergence criteria:

  parameters = 3.833e-05 likelihood = 0.0002426 gradient = 1.137e-06
```

```
summary(joint.surro.sim.MCPGH)
  Estimates for variance parameters of random effects
          Estimate  Std Error       z          P
  theta       2.640    0.4295    6.148  7.854e-10 ***
  zeta        2.277    0.4010    5.679  1.356e-08 ***
  gamma       1.355    0.4174    3.246    0.00117  **
  alpha       1.135    0.2285    4.965  6.855e-07 ***
  sigma2_S    0.593    0.3471    1.709    0.0875   .
  sigma2_T    0.664    0.5771    1.151    0.2498
  sigma_ST    0.380    0.3219    1.181    0.2376
  Estimates for fixed treatment effects
        Estimate Std Error      z          P
  beta_S   -1.643    0.2277 -7.216    < e-10 ***
  beta_T   -1.640    0.3573 -4.589  4.463e-06 ***
  ---
  Signif. codes: 0 '***' 0.001 '**' 0.01 '*' 0.05 '.' 0.1 ' ' 1
  hazard ratios (HR) and confidence intervals for fixed treatment effects
        exp(coef) Inf.95.CI Sup.95.CI
  beta_S     0.193     0.124     0.302
  beta_T     0.194     0.096     0.391
  Surrogacy evaluation criterion
            Level Estimate Std Error Inf.95.CI Sup.95.CI Strength
  Ktau    Individual   0.577        --     0.522     0.607
  R2trial      Trial   0.367     0.358    -0.334     1.068      Low
  R2.boot      Trial   0.407        --     0.007     0.964      Low
  ---
  Association strength: <= 0.49 'Low';]0.49-0.72['Medium'; >= 0.72 'High'
  ---
  Surrogate threshold effect (STE): -4.922 (HR = 0.007)
  Convergence parameters
```

```
Penalized marginal log-likelihood = -4968.465

Number of iterations = 20

LCV = the approximate likelihood cross-validation criterion
      in the semi-parametrical case = 8.317

Convergence criteria:
  parameters = 5.962e-05 likelihood = 0.0004484 gradient = 2.465e-06
```

**Graphical representation of baseline hazard and survival functions.** By using the generic function `plot()`, it is possible to plot the baseline hazard and survival functions for both surrogate and true endpoints. The definition of this function is shown below, and the associated arguments are described in S2E Appendix in S2 Appendix.

```
plot(x, endpoint = 2, scale = 1, type.plot = "Hazard",
xmin = 0,
    conf.bands = TRUE, xmax = NULL, ylim = c(0, 1), Xlab =
"Time",
    pos.legend = "topright", main, cex.legend = 0.7,
    Ylab = "Baseline hazard function")
```

Fig 1 represents the baseline survival and hazard functions for model, for both the surrogate and the true endpoints using the advanced ovarian cancer meta-analysis dataset. We limit survival times to 8 months since after this threshold, the estimated survival probabilities are almost equal to 0. The code below produces the plots given in Fig 1.

```
par(mfrow = c(2, 1))

plot(joint.surro.ovar,type.plot = "Su", xmax = 8, Xlab =
"Time (in months)",
    scale = 12)

plot(joint.surro.ovar, xmax = 8, ylim = c(0, 0.2), Xlab =
"Time (in months)",
    scale = 12, pos.legend = "topleft")
```

Fig 2 shows another representation of the baseline survival and hazard functions for the surrogate and the true endpoints. We use the object `joint.surro.sim.MCPGH` for this purpose, which is based on the generated data.

The following code is used to produces Fig 2:

```
par(mfrow = c(2, 2))

plot(joint.surro.sim.MCPGH, type.plot = "Su", endpoint = 0,
scale = 1/365,
    Xlab = "Time (in years)")

plot(joint.surro.sim.MCPGH, type.plot = "Su", endpoint = 1,
scale = 1/365
```

```
        ,pos.legend = "bottomleft", Xlab = "Time (in years)")

plot(joint.surro.sim.MCPGH, type.plot = "Ha", endpoint = 0,
scale = 1/365

    ,ylim = c(0, 0.08),

    Xlab = "Time (in years)")

plot(joint.surro.sim.MCPGH, type.plot = "Ha", endpoint = 1,
scale = 1/365

    ,ylim = c(0, 0.08),

    Xlab = "Time (in years)")
```

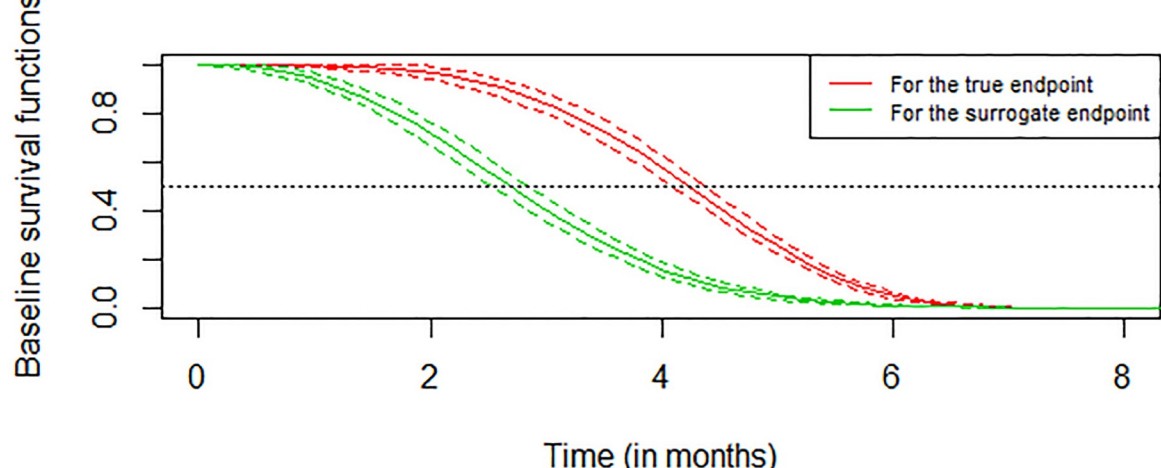

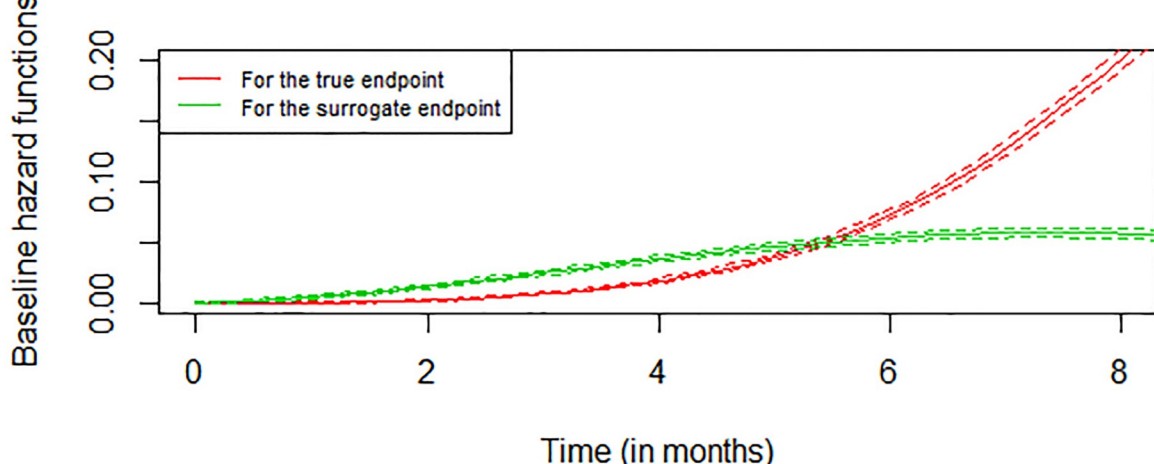

**Fig 1. Baseline hazard and survival functions for surrogate endpoint and true endpoint truncated at 8 months using the advanced ovarian cancer meta-analysis dataset.**

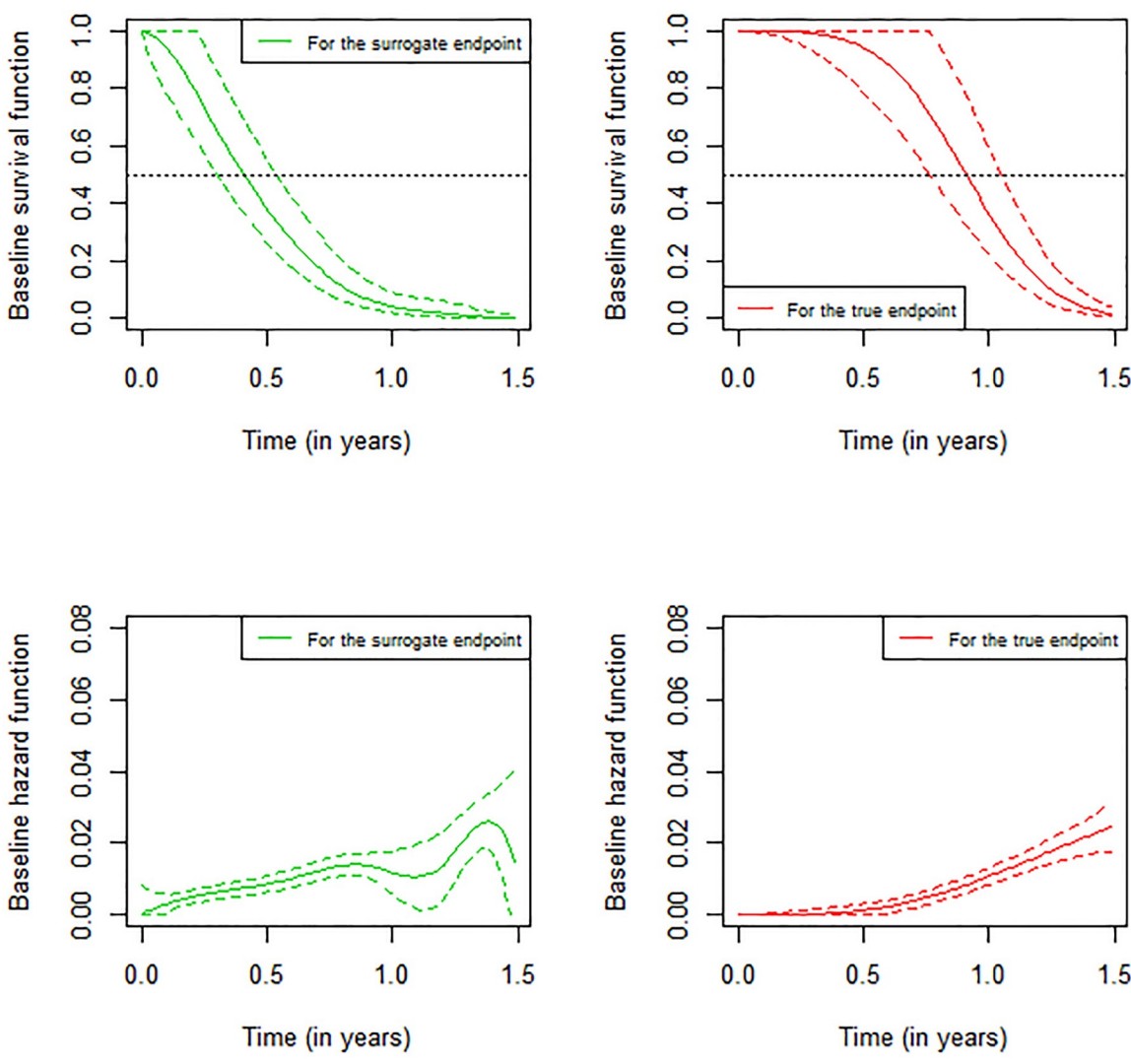

**Fig 2. Baseline hazard and survival functions for surrogate endpoint and true endpoint, using simulated meta-analysis of 600 subjects and 30 trials.**

## Model evaluation and prediction

To assess the accuracy of the prediction using estimates from model (1), the leave-one-out cross validation criteria (loocv) described in S2F Appendix in S2 Appendix can be performed as follows:

```
dloocv <- loocv(object = joint.surro.sim.MCGH, unusedtrial =
26,

var.used = "error.estim")
```

We found the following result:

```
dloocv$result
  trialID ntrial    beta.S    beta.T   beta.T.i  Inf.95.CI  Sup.95.CI
1       1     20    -2.145    -0.582    -2.038    -2.663    -1.412
2       2     20    -1.480    -0.799    -1.464    -2.135    -0.793  *
3       3     20    -0.285    -0.422    -0.195    -1.801     1.411  *
4       4     20     0.307     0.487    -0.248    -2.347     1.852  *
5       5     20    -1.087    -1.230    -0.983    -2.007     0.041  *
6       6     20   -21.305    -1.496   -13.951   -32.636     4.733  *
7       7     20    -0.796    -1.943    -0.687    -1.889     0.515
8       8     20    -1.578    -1.302    -1.545    -2.167    -0.923  *
9       9     20    -1.909    -1.402    -1.736    -2.241    -1.230  *
10     10     20    -1.752    -0.053    -1.505    -2.174    -0.836
11     11     20   -21.304    -0.342   -16.325   -35.269     2.619  *
12     12     20    -2.766   -20.920    -2.236    -3.201    -1.271
13     13     20    -0.474    -1.025    -0.835    -2.289     0.618  *
14     14     20     0.056    -0.148    -0.561    -2.603     1.481  *
15     15     20    -1.337    -1.154    -1.250    -2.218    -0.282  *
16     16     20    -0.191    -0.291    -0.125    -1.833     1.582  *
17     17     20     0.264     0.161    -0.540    -3.006     1.926  *
18     18     20    -2.589    -0.657    -2.197    -2.968    -1.426
19     19     20    -1.795    -1.654    -1.562    -2.263    -0.861  *
20     20     20    -0.630     1.599    -1.128    -2.451     0.195
21     21     20    -0.593    -0.510    -0.602    -1.988     0.785  *
22     22     20    -0.682    -1.645    -0.555    -1.827     0.716  *
23     23     20    -0.787    -0.179    -0.850    -2.061     0.362  *
24     24     20    -3.019    -2.735    -2.227    -3.504    -0.949  *
25     25     20    -2.393    -1.577    -2.099    -2.879    -1.319  *
26     27     20    -1.640    -1.063    -1.630    -2.248    -1.012  *
27     28     20    -1.386    -1.672    -1.220    -2.057    -0.383  *
28     29     20    -0.207    -0.722    -0.535    -2.220     1.150  *
29     30     20     0.299     0.185    -0.377    -3.215     2.461  *
```

The returned object, of class `jointSurroPenalloocv` includes for each trial the number of included subjects (`ntrial`), the observed treatment effect on the surrogate endpoint (`beta.S`), the observed treatment effect on the true endpoint (`beta.T`) and the predicted treatment effect on the true endpoint (`beta.T.i`) with the associated prediction interval (`Inf.95.CI`, `Sup.95.CI`). If the observed treatment effect on the true endpoint is included into the prediction interval, the last column contains "*", indicating a good prediction.

## Simulation studies

In this section, we show an example of simulation studies in the `frailtypack` package, based on model (1).

**Estimations.** Using the function `jointSurroPenalSimul()` simulation studies can be performed as follows:

```
joint.simul10 <- jointSurroPenalSimul(nb.dataset = 10,
nbSubSimul = 600,

    ntrialSimul = 30, LIMparam = 0.001, LIMlogl = 0.001,
LIMderiv = 0.001,

    nb.mc = 200, nb.gh = 20, nb.gh2 = 32, true.init.val = 1,
print.iter = F)
```

This function serves to perform simulation studies with 10 meta-analyses, each study including 600 subjects and 30 trials. By default, each generated meta-analysis includes the same proportion of subjects per trial and the same proportion of treated subjects per trial. In the event of an identification problem, the model is re-estimated using 32 quadrature nodes. All unused simulation parameters are set to the initial value, as presented in the function `jointSurroPenalSimul()`. Using default values, we expect $0.81$ for $R^2_{trial}$ and $0.595$ for Kendall's $\tau$.

**Simulation results.** Simulation results can be displayed using the S3 method `summary()`. This function allows argument `R2boot` to specify whether the confidence interval of $R^2_{trial}$ will be computed using parametric bootstrapping (`1`) or the Delta method (`0`).

```
summary(joint.simul10, R2boot = 0)

  Simulation and estimation pamareters

  nb.subject = 600

  nb.trials = 30

  nb.simul = 10

  int.method = 2

  nb.gh = 20

  nb.gh2 = 32

  nb.mc = 200

  kappa.use = 4
```

```
n.knots = 6

n.iter = 14

Simulation results

  Parameters True value   Mean Empirical SE Mean SE CP(%)
2      theta        3.5  3.451      0.711   0.545    80
3       zeta          1  1.049       0.22   0.177    70
4      gamma        2.5  2.642      0.957   0.711    80
5      alpha          1  1.009      0.135   0.138    90
6    sigma.S        0.7  0.608      0.361   0.426    90
7    sigma.T        0.7  0.627      0.347   0.459    80
8   sigma.ST       0.63  0.555      0.314   0.389    90
9     beta.S      -1.25 -1.368      0.233   0.251    90
10    beta.T      -1.25 -1.397      0.238   0.269   100
11   R2trial       0.81   0.82      0.181   0.521    80
12     K.tau      0.595  0.592      0.032       -    80
Rejected datasets: n(%) = 0(0)
```

In the first part of the results, we present a brief summary of simulation and estimation parameters, and the average number of iterations to reach convergence (`n.iter = 14`).

The next part presents a table of simulation results. Each row of the table corresponds to a model parameter. The first column is the name of the parameter, followed by the value assigned to the parameter during simulation. The next three columns correspond to the average of the estimates observed for all the generated datasets, the empirical standard errors and the mean of the estimated standard error. The last column is the coverage probability (CP), which is the proportion (%) of the 95% confidence intervals of the estimate that includes the true value of the parameter. We considered 10 meta-analyses here, although simulation studies more often require around 500 datasets of meta-analysis.

The last row of the results indicates the number of rejected datasets due to convergence issues.

## Discussion

This paper presents new tools for validating candidate surrogate endpoints using data from multiple randomized clinical trials, with failure time endpoints. Since version `3.0.1`, The `R` `frailtypack` package implements the joint-surrogate model, which is a more attractive approach than two-step approaches for evaluating surrogate endpoints based on a one-step analysis strategy. The joint-surrogate model demonstrated better performances than the two-step copula model or the one-step Poisson approach [17]. Furthermore, the new model showed stable results even with a moderate trial size or number of trial as commonly encountered in practice, whereas the adjusted model estimated with the Bayesian framework showed unstable results [11]

By varying the values of the arguments in the `jointSurroPenal` function, convergence of the model is not always guaranteed. Therefore, it is important in the event of convergence issues to know how to play with the arguments/values couple as shown in the previous section. Thus, users can choose the method of integration, initial values, the number of nodes for splines and the smoothing parameters, the number of nodes to use for the Gauss-Hermite quadrature and the number of samples for the Monte-Carlo integration when applicable, the random number generator, and other necessary arguments. It is also possible to set some parameters of the model in the event of identifiability issues. This underlines the flexibility of the `frailtypack` package in managing convergence issues. This flexibility is quite different from that obtained with the `surrosurv` package [9] or macros `SAS` [10] for evaluating surrogate endpoints using the two-step Copula model or one-step Poisson model. Other advantages of our model compared to existing approaches [8, 13] are in the reduction of convergences and numerical issues, the robustness to model misspecification, the surrogacy evaluation based on a one-step approach and therefore the estimation of $R^2_{trial}$ without need for adjustment on estimation errors. In addition, as underlined in the illustration section, the interpretation of Kendall's $\tau$ is different from that in the two-step copula approach.

Our previous paper [17] demonstrated the robustness of the joint surrogate model to model misspecification, numerical integration and variations in data characteristics regarding the surrogacy evaluation criteria ($R^2_{trial}$ and Kendall's $\tau$). It is robust to variations in the values of the arguments regarding the surrogacy evaluation criteria. Thus, in the event of convergence, change in arguments/values mostly produced similar results. For example, when we reduced the number of samples for Monte-Carlo integration to `100 (nb.mc = 100)` in the application based on the advanced ovarian cancer meta-analysis dataset, we observed `R2trial = 1.000 [95%CI: 0.998-1.002), R2boot = 0.981 [95%CI: 0.891-1.000], Kendall's` $\tau$ `= 0.683 [95%CI: 0.664-0.695],STE = -0.291 (HR = 0.747)` and `LCV = 9.161`. These results are quite similar to those using `nb.mc = 200` (see illustration section in manuscript). In addition, if we integrate over the random effect at the individual level using the pseudo-adaptive Gaussian-Hermite quadrature (argument `adaptatif = 1`) instead of the classical Gaussian-Hermite quadrature, the results are similar with `R2trial = 1.000 [95%CI: 0.998-1.002], R2boot = 0.982 [95%CI: 0.897-1.000],Kendall's` $\tau$ `= 0.683 [95%CI: 0.664-0.696], STE = -0.272 (HR = 0.762)` and `LCV = 9.162`. These examples confirm the robustness of the model previously discussed by Sofeu et al. (2019) using simulation studies.

Moreover, thanks to the `jointSurroPenalSimul()` function, it is possible to perform simulation studies in order to plan a new trial and define the optimal number of clusters when evaluating surrogate endpoints given the joint surrogate model. For example, if a given meta-analysis includes few trials, simulation studies may help in establishing the minimum number of centers to obtain the best estimate of the surrogacy evaluation criteria. Jurgen et al. [33] suggested using clinical trial simulations to optimize adaptive trial designs. As they explained, the typical goal of a clinical trial simulation is to identify a design that has a high probability of success based on the most likely conditions but which can also perform well, or at least acceptably, under more extreme conditions if necessary. Simulation studies can help if the recommended values for the arguments do not make it possible to reach convergence or involve longer computer time when fitting the joint surrogate model. Given the data characteristics, they can help in choosing optimal values for some arguments (the number of quadrature nodes, the number of samples for the Monte-Carlo integration and the number of nodes for splines) and in anticipating their impact on estimating the model parameters. The management of the convergence issues by the program itself is described in S2G Appendix in S2 Appendix.

Numerous tools have been presented in this paper for evaluating surrogacy. We have the following: the surrogate threshold effect which is used in combination with $R^2_{trial}$ to assess the validity of the potential surrogate endpoint; the `predict()` function used in a new trial to predict the treatment effect of the true endpoint based on the observed treatment effect on the surrogate endpoint; and the leave-one-out cross-validation which can be used to assess the accuracy of the prediction using model (1). Furthermore, a graphical representation of the baseline hazard and survival functions is possible using the `plot()` function.

The `jointSurroPenal()` function can also be used in interim analyses to estimate the fixed treatment effect on the surrogate endpoint, taking into account competing risk of death and heterogeneity in the data at the individual level and at the trial level in interaction with treatment. This is an alternative to the joint frailty-copula model between tumor progression and death for meta-analysis proposed in [34].

We now plan to extend the model (1) and the `jointSurroPenal()` function to take into account interval censoring for endpoints where the exact event times are unknown. This extension will also make it possible to model the baseline hazard functions parametrically, using a Weibull distribution. To improve the use of `frailtypack`, intuition can be gained by developing an associated interactive web app using the R package `Shiny` available at https://CRAN.R-project.org/package=Shiny.

## Supporting information

**S1 Fig. Package characteristics (version 3.0.3.1).** Blue cross is for the option available for a given type of model in the package on CRAN, orange cross is for the option included in the package but not yet on CRAN yet. Empty cells mean that an option is not available for a given type of model. RE = Recurrent Event. TE = Terminal Event. LO = Longitudinal Outcome. STE = Surrogate Threshold Effect. ODE = Ordinary Differential Equation.
(TIF)

**S1 Appendix. Extension of the methodology.**
(PDF)

**S2 Appendix. Description of the arguments and return values for the functions.**
(PDF)

## Acknowledgments

The authors thank the Ovarian Cancer Meta-Analysis Project for sharing the data used to illustrate the programs. This work was supported by the Association pour la Recherche sur le Cancer, Grant/Award Number: PJA20161205147; Institut National du Cancer, Grant/Award Number: 2017-125; Institut national de la santé et de la recherche médicale; Région Aquitaine. We also thank Antoine Barbieri, INSERM U1219, for his support for programming the Bayesian approach. We gratefully acknowledge very helpful and constructive comments and suggestions from the academic editor and the three anonymous referees, which lead to significant improvements of this manuscript.

## Author Contributions

**Conceptualization:** Casimir Ledoux Sofeu, Virginie Rondeau.

**Data curation:** Casimir Ledoux Sofeu.

**Formal analysis:** Casimir Ledoux Sofeu.

**Funding acquisition:** Virginie Rondeau.

**Methodology:** Casimir Ledoux Sofeu, Virginie Rondeau.

**Project administration:** Virginie Rondeau.

**Software:** Casimir Ledoux Sofeu.

**Supervision:** Virginie Rondeau.

**Validation:** Casimir Ledoux Sofeu.

**Visualization:** Casimir Ledoux Sofeu.

**Writing – original draft:** Casimir Ledoux Sofeu.

**Writing – review & editing:** Casimir Ledoux Sofeu, Virginie Rondeau.

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
