## [Decision Letter · Decision Letter 0]

11 Sep 2019

PONE-D-19-20769

Frailtypack: an R-package for the validation of failure-time surrogate endpoints using individual patient data from meta-analysis of randomized controlled trials

PLOS ONE

Dear Mr. SOFEU,

Thank you for submitting your manuscript to PLOS ONE. After careful consideration, we feel that it has merit but does not fully meet PLOS ONE’s publication criteria as it currently stands. Therefore, we invite you to submit a revised version of the manuscript that addresses the points raised during the review process.

We would appreciate receiving your revised manuscript by Oct 26 2019 11:59PM. To enhance the reproducibility of your results, we recommend that if applicable you deposit your laboratory protocols in protocols.io, where a protocol can be assigned its own identifier (DOI) such that it can be cited independently in the future. For instructions see: http://journals.plos.org/plosone/s/submission-guidelines#loc-laboratory-protocols

We look forward to receiving your revised manuscript.

Kind regards,

Alan D Hutson

Academic Editor

PLOS ONE

Journal Requirements:

Additional Editor Comments (if provided):

There were three well thought-out reviews for this submission. Please address the important points put forward by each reviewer.

Reviewers' comments:

Reviewer's Responses to Questions

**Comments to the Author**

1. Is the manuscript technically sound, and do the data support the conclusions?

Reviewer #1: Partly

Reviewer #2: Yes

Reviewer #3: Partly

2. Has the statistical analysis been performed appropriately and rigorously? 

Reviewer #1: N/A

Reviewer #2: Yes

Reviewer #3: Yes

3. Have the authors made all data underlying the findings in their manuscript fully available?

Reviewer #1: Yes

Reviewer #2: Yes

Reviewer #3: Yes

4. Is the manuscript presented in an intelligible fashion and written in standard English?

Reviewer #1: No

Reviewer #2: No

Reviewer #3: Yes

5. Review Comments to the Author

Reviewer #1: The manuscript by Sofeu and Rondeau covers an interesting topic and a relevant research question, and I think it warrants publication. However, I think some changes are needed to the manuscript.

For instance, I think it would be a much stronger paper if the application of the joint frailty model described in this paper was expanded and described in more detail, to motivate the use of the newly developed method; I would also suggest removing the simulated data and focussing on the ovarian cancer dataset.

The submitted manuscript is quite technical, focussing on describing the software package rather than the practical importance of it. I think it would appeal to a wider audience if contextualised more.

Regarding the applied examples, I think that the choice of arguments (e.g. numerical integration method, number of quadrature nodes, etc.) should be discussed in more detail, as it can significantly affect the results (and not all users are aware of these issues). I would also recommend discussing the robustness of the method to model misspecification, and the consequences of varying the estimation arguments mentioned above.

Finally, the authors mention several alternative methods to assess surrogacy; their comparison with the joint frailty model should be discussed in more detail (and maybe it would be interesting to include them in the application section as well, for comparison purposes).

More comments are included below.

Introduction:

I think that the baseline hazard function that uses splines should be referred to as “parametric” (or “flexible parametric”), rather than “non-parametric”. Survival models with flexible, spline-based baseline hazards are commonly referred to as "flexible parametric models".

Parameters estimation:

Given that several numerical integration methods have been considered, what has been chosen and why? How does the choice of integration method affect the validity of the method?

STE:

What is IQWIG?

Computational details:

Not all researchers can afford a Xeon with 40 cores and 300+ Gb of RAM. Could you elaborate on the computational requirements of this method? Would it be possible to fit any model on e.g. a standard laptop or desktop PC?

You are using R 3.4.3, which was released almost 2 years ago. How does the software run with newer versions of R?

You mention that dependencies must be installed. Could you describe them, to make the readers aware?

Data:

Minor comment: data(“dataOvarian”) only works if frailtypack is loaded first; you could add the “package” argument to make the requirement explicit: data(“dataOvarian”, package = “frailtypack”)

Surrogacy evaluation:

The arguments of “jointSurroPenal” that were set when fitting the joint frailty model with the ovarian cancer dataset are described but not motivated. Why were those specific values chosen? Does this affect the results of the joint model? If so, how? Model misspecification is a serious issue that can lead to biased estimates of the treatment effect, also in frailty models.

Choice of the model based on LCV:

To me, the two models don’t give similar results. For instance, the estimated fixed treatments effects are quite different (e.g. -2 vs -1.6 for beta_S). This ties with some of my previous comments on model misspecification and the choice of the estimation arguments when fitting the joint frailty model.

Simulation studies:

I don’t understand the utility of this section. Could you elaborate a bit more on that? If you simulate data from a joint frailty model and then fit a joint frailty model with the same model formulation to the simulated data, then good performance is expected. I am probably missing the point here (sorry for that), but it would be good to describe in more detail why and when this is useful.

Discussion:

The possibility of choosing estimation arguments that affect convergence of the algorithm should be discussed more, including comparisons (e.g. numerical integration methods), drawbacks, and problems that may arise.

A comparison with other established methods to assess surrogacy should be discussed as well, motivating the use of joint frailty models. Furthermore, robustness of the software introduced with this manuscript should be discussed.

Appendices:

There is a lot of material in the appendices - is it all necessary? I believe some of it is already included in the paper introducing the methodology (Sofeu et al., 2019, in Statistics in Medicine), and maybe readers could be referred to that instead.

Finally, the paper is at times hard to read, with some typos and several sentences that could be edited to improve clarity (remember that PLOS ONE does not copyedit accepted manuscripts). I spotted some typos, which are included below:

Title: “frailtypack” should start with a lowercase letter since it is the name of the package;

Keywords: “fraity” instead of “frailty”; “surrogte” instead of “surrogate”; “envent” instead of “event”

Abstract: The first sentence is hard to read; “quiet” instead of “quite” in the background and objective section (line 9);

Line 89: replace “discuss” with “discussed”

Line 131: replace “measurements” with “measure”

Line 134: replace “interpreting” with “interpretation”

Line 136: replace “equals” with “equal”

Line 225: replace “describe” with “described”

Line 237: GitHub should have capitals G and H (it’s the name of the company/service)

Line 241: did you mean Gb instead of Go?

Line 342: replace “respectievely” with “respectively”

Line 365: spell out “loocv”

Line 447: replace “censorship” with “censoring”

Line 448: replace “which” with “where”

Reviewer #2: Sofeu et.al. presented a manuscript that is an R implemented of their method published in “One-step validation method for surrogate endpoints using data from multiple randomized cancer clinical trials with failure-time endpoints. Statistics in Medicine. 2019;1–15”, with some enhanced functions for the validation of candidate surrogate endpoints. The manuscript presented some potentially useful alternative method for validate surrogate endpoint, based on the method published in the associated Statistics in Medicine paper. This package could potentially help researchers to choose more appropriate statistical method when validate surrogate endpoints. However, there is lack of demonstration of the stated advantages in comparison to other methods, and detailed discussion of the potential benefit of using the stated method is missing. Only general comparison was presented in the “Discussion” section without much depth or details. As such, the stated advantages were not well established in this manuscript. Overall, the manuscript mostly demonstrated the technical side of the package. Therefore, this reviewer encourages the author to consider enhances the manuscript so a researcher (instead of a package user) can benefit from reading the manuscript and decided how to realize its potential statistical advantages when used appropriately.

Major comments:

1. Numerous English language grammar errors in the manuscript is a real problem. Some of them listed later in the minor comments. Many sentences are nearly incomprehensible. An English proof reader is in need.

2. In the introduction section, the authors briefly discussed the pros and cons of existing methods and their method for surrogate endpoint validation. Yet, the manuscript (and package) focused on meta-analysis of such data. It might be obvious to the authors why such data is well suit and/or needed, but it should be make clear that a) if the method can be used for single study/center data, and b) the rational of using meta-analysis data in the demonstration of the manuscript. This is vital for the readers to understand the applicability and limitations of their package/methods.

3. In the associated R package, the Vignettes did not provide enough details for even the minimal example. On the other hand, this manuscript provided only marginally more information than a package Vignettes. Some statistical insights and details would make this manuscript more useful for a researcher instead of an R package user, for example, practical considerations in model estimation, interpretation of the results (like the summary presented in page 12-13), and any potential issues when choose different parameters. In addition, any practical advantages (if any) derived from these results using the method for the Ovarian Cancer data presented here would be very useful in demonstrate the benefit of this method over others. If not yet available, at least discuss potential gains by using this method instead of the others.

4. It is not clear what the results in the “Simulation studies” section can be used in the context of the Ovarian Cancer data analysis, other than a brief sentence in the Discussion section. The author may want to explain how such results are useful for the stated purpose (plan future trial) there.

Minor comments:

1. In Abstract the sentence “We have especially the surrogate threshold effect…” need revision to make a valid English sentence. Also “Other tools concerned data generation, studies simulation and graphic representations…”.

2. Many minor grammar issues in main text.

Reviewer #3: This paper addresses an interesting problem, one which is numerically challenging to solve. I have comments concerning the presentation, the statistical problem, and the software. The last of these three is separated out and should be viewed as comments on possible future evolution of the package, but is not relevant to any editorial decisions about the current manuscript, which deals with the present offering of the package.

1. I found the Introduction confusing. The title of the paper leads one to believe that this is a description of a particular package, it needs an early lead in to the fact that this is NOT what the paper is about. Two or three sentences would do, e.g., "frailtypack is an R package that fits a variety of

survival models containing one or more random effects, or frailties...." "it includes for instance simple shared frailty, correlated...." "...for this paper we will focus on a particular subset of features applicable to the evaluation of surrogate endpoints ..." We need just enough to orient the user so that they don't feel like they were dropped into chapter 2 of a novel.

2. The description of the competing methods, starting at about page 2 line 14, was quite difficult to follow. The paragraphs need "roadmap" sentences to help readers who have not immersed themselves in the field know what the goal of the journey (paragraph) will be. It is hard to know what is background and what is essential. (This reviewer for instance -- I know some of the overarching questions but it is not a personal research area.) Without that the information is both too much and too little.

(page 3, lines 78-85 are nicely done.)

3. Equation 1 is surprising -- where are the covariates? (See line 17, which introduced a need for them.) I realize that in practical work there will be at most 1 or 2 prognostic factors that are widely enough recognized such that all studies will have gathered them. Can you sidestep the issue by creating extra strata?

When Z is a 0/1 covariate, as is often the case, then $\\v_S$ and $v_T$ are not identifiable for the control subjects. Does this cause issues wrt estimation of the variance of $v$?

4. The function on lines 163 and following needs some explanation. First there is no formula. Only later when reading the examples, was this reviewer able to guess that you decided that the user should use a certain set of predetermined variable names. Please add some text.

(Software suggestion: There list of options is terribly long; how is one to know what is necessary and what is optional. The pattern found in glm.control and many other packages would be a good way to separate out the secondary ones.)

5. The double hash signs on the left of the printout are a distraction and should be removed. (When there is lots of code, little output, and the goal is to allow users to easily cut/paste, then this peculiar design choice is defensible. None of those 3 is true here.)

6. The discussion of the results on pages 10-11 is an important part of the paper. The authors have several good choices in the material just prior to this section wrt moving significant portions of it to the appendix. However, several details of the discussion are very terse. For example the statements at lines 298-302. Why is a value of 6.8 "strong" and a value of 1.79 "more pronounced".

7. A first question of an estimation method is whether it works when the data set exactly fits the proposed model (if that fails then an approach is nearly useless). The far more important one is how the method works when the model is not quite true, i.e., the case for any real data set. The package's simulation modules are designed to address only the first of these, not the second, which is a serious limitation. More serious is the lack of any evaluation of the approach outside of the ideal case. Some methods are methods are robust to such changes and some are fragile.

8. Figure 1 is not very helpful, and is in fact confusing. Arrows normally go from parent to child. Do you really mean to imply that the plot() function generates data that is used by the estimation functions? Why is there an arrow out the side of summary()?

Figures 2 and 3 are at odds. The paper states that they are two "representations" of a fit to the same data, yet Figure 2 has median values of 2.5 and 4 months, while figure 3 shows values close to .5 and 1 year. If these are simply two different solution paths, it would argue that the software is very unstable. No discussion is provided to guide the user with respect to these discrepant results.

Minor comments (no response required)

Abstract sentence 1: " ... for accelerated effectively the phase 3 trial." I've read this 3 times and still do not know what the words mean.

Abstract: "This model was quiet robust..." I think you mean 'quite robust'.

$\\alpha$ and $\\zeta$ might make a little more sense attached to S rather than T. After all, death is death, but different institutions can and will have different standards for "progression".

6. PLOS authors have the option to publish the peer review history of their article (what does this mean?). If published, this will include your full peer review and any attached files.

Reviewer #1: No

Reviewer #2: No

Reviewer #3: No

---

## [Author Response · Author response to Decision Letter 0]

29 Oct 2019

We thank the editor and the reviewers for the great interest to our manuscript. Following the comments of the editor, about the Plos One’s style requirements, we numbered the body of the manuscript from the Author list and use plos One’s Latex template to write the manuscript. 

According to the editor and the reviewers’ comments, the manuscript has been copyedited by a professional scientific copyeditor: 

RAYMOND COOKE 

20 Rue Louis MONDAUT

33150 CENON

France

Please find below our replies to the reviewers’ comments. We have also revised the manuscript in accordance with the comments when necessary.

5. Review Comments to the Author

Reviewer #1: The manuscript by Sofeu and Rondeau covers an interesting topic and a relevant research question, and I think it warrants publication. However, I think some changes are needed to the manuscript.

1- For instance, I think it would be a much stronger paper if the application of the joint frailty model described in this paper was expanded and described in more detail, to motivate the use of the newly developed method; I would also suggest removing the simulated data and focussing on the ovarian cancer dataset.

The submitted manuscript is quite technical, focussing on describing the software package rather than the practical importance of it. I think it would appeal to a wider audience if contextualised more.

We thank reviewer #1 for this remark. The paper describes the use of a new software package for evaluating surrogate endpoints, rather than the methodological aspects already published in Statistics in Medicine (Sofeu et al., 2019). We take your point that we could emphasize the practical importance of it. To this end, we now discuss the choice of arguments with the associated values, the management of convergence issues and the interpretation of outputs, rather than just illustrate the use of the functions.

To improve the illustration of the joint surrogate model and then guide the choice of the values for the arguments of the jointSurroPenal() function, we illustrate the call of this function by adding the following paragraphs in the manuscript (see the illustration section): 

“From a practical point of view, the most important arguments for using the jointSurroPenal() function beyond the standard argument (data) concern the following: the parametrization of the model (with arguments indicator.zeta and indicator.alpha), the method of integration and associated arguments (int.method, n.knots, nb.mc, nb.gh, nb.gh2, adaptatif), the smoothing parameters (init.kappa and kappa.use) and the scale of survival times (scale). Although optional, all these arguments can be used to manage the convergence issues. The choice of the values to assign to these arguments can be based on the convergence of model. When the convergence issues are fixed, users can implement the likelihood cross-validation criteria to evaluate the goodness of fit of different models, as shown later in this section. In the first step, users can try the model with the default values. 

In the event of convergence issues, we recommend the following strategy: changing the number of samples for Monte-Carlo integration (nb.mc) by choosing a numerical value between 100 and 300; varying the number of nodes for the Gaussian-Hermite quadrature integration (nb.gh and nb.gh2) by choosing the values between 15, 20 and 32 varying the number of nodes for spline (n.knots) by a numerical value between 6 and 10; providing new values for the smoothing parameters (init.kappa). Users can also set the arguments alpha or zeta to 1 (indicator.zeta = 1 or indicator.alpha = 1) to avoid estimating these parameters. We also recommend changing the integration method with the arguments int.method and adaptatif. For example, by using adaptatif = 1 for integration over the random effects at the individual level, one could use the pseudo adaptive quadrature Gaussian-Hermite integration instead of the classical quadrature Gaussian-Hermite method. By changing the scale of the survival times (argument scale) and considering years instead of days, it is possible to solve some of the numerical issues. 

Using the default values based on the advanced ovarian cancer dataset, the model did not converge. By changing the value of some arguments, we obtained the following set of arguments/values which allowed it to converge: “

We agree that the core of the application should be concentrated around the use of the joint surrogate model based on real data. This is why we emphasized the choice of argument/values and the interpretation of output from the jointSurroPenal() function on real data. However, simulated data were used owing to the description of the jointSurrSimul() functions for generating a meta-analysis dataset and then to illustrate the simple use of different functions and their arguments for surrogacy evaluation. 

2- Regarding the applied examples, I think that the choice of arguments (e.g. numerical integration method, number of quadrature nodes, etc.) should be discussed in more detail, as it can significantly affect the results (and not all users are aware of these issues). I would also recommend discussing the robustness of the method to model misspecification, and the consequences of varying the estimation arguments mentioned above.

We thank the reviewer for this comment. The choice of arguments/values and management of arguments in the event of convergence issues are now discussed in detail. We now include a comment on this point (see response comment #1). 

Concerning the robustness of the method, it was detailed in the previous methodological paper (Sofeu et al. 2019) based on simulation studies. The authors showed that the joint surrogate model is quite robust to model misspecification, numerical integration and variations in data characteristics regarding the surrogacy evaluation criteria (R2trial and Kendall’s Tau). Therefore, in the event of convergence of the model, the parameters are generally good estimates. We have added the following comment in the discussion section:

“Our previous paper (Sofeu et al., 2019) demonstrated the robustness of the joint surrogate model to model misspecification, numerical integration and variations in data characteristics regarding the surrogacy evaluation criteria (R2trial and Kendall’s Tau). 

In the following, we discuss the robustness of the model regarding variations in the values of some arguments based on the advanced ovarian cancer meta-analysis dataset. We then include the following paragraph in the revised discussion section:

“In addition, we demonstrate the robustness of the model to variations in the values of the arguments regarding the surrogacy evaluation criteria. Thus, in the event of convergence, changes in argument/value mostly produced similar results. For example, when we reduced the number of samples for Monte-Carlo integration to 100 (nb.mc = 100) in the application based on the advanced ovarian cancer meta-analysis dataset, we observed R2trial = 1.000 [95%CI: 0.998 – 1.002), R2boot = 0.981 [95%CI: 0.891 – 1.000], Kendall’s Tau = 0.683 [95%CI: 0.664 – 0.695], STE = -0.291 (HR = 0.747) and LCV = 9.161. These results are quite similar to those using nb.mc = 200 (see illustration section in manuscript). In addition, if we integrate over the random effect at the individual level using the pseudo-adaptive Gaussian-Hermite quadrature (argument adaptatif = 1) instead of the classical Gaussian-Hermite quadrature, the results are similar with R2trial = 1.000 [95%CI: 0.998 – 1.002], R2boot = 0.982 [95%CI: 0.897 – 1.000], Kendall’s Tau = 0.683 [95%CI: 0.664 – 0.696], STE = -0.272 (HR = 0.762) and LCV = 9.162. These examples confirm the robustness of the model previously discussed by Sofeu et al. (2019) using simulation studies.”

3- Finally, the authors mention several alternative methods to assess surrogacy; their comparison with the joint frailty model should be discussed in more detail (and maybe it would be interesting to include them in the application section as well, for comparison purposes).

In the methodological article published in Statistics in Medicine (Sofeu et al., 2019), we already compared the joint surrogate model with the two-step copula approach of Burzykowski et al. (2001) and with the one-step Poisson approach of Rotolo et al. (2017). The joint surrogate model was quite robust to the misspecification of data and variations in data characteristics compared to existing approaches. In addition, convergence and identifiability issues were attenuated with the new model. These findings were detailed by the authors. Therefore, to avoid dual publication and given that the ongoing manuscript aims to reach out to a wide community of users, we mainly focus on the description and usage of the package, with additional tools for evaluating surrogacy. 

More comments are included below.

Introduction:

4- I think that the baseline hazard function that uses splines should be referred to as “parametric” (or “flexible parametric”), rather than “non-parametric”. Survival models with flexible, spline-based baseline hazards are commonly referred to as "flexible parametric models".

We thank the reviewer for this remark. We have corrected the mistake by replacing the expression “non-parametric” by “flexible” in the manuscript for the baseline hazard function. However, the model parameters and baseline hazard functions were estimated using a semi-parametric penalized likelihood approach. 

Parameters estimation:

5- Given that several numerical integration methods have been considered, what has been chosen and why? How does the choice of integration method affect the validity of the method?

The choice of the numerical integration method is governed by the convergence of the model and the computational time. As a first step, we encourage users to choose a combination of Monte-Carlo to integrate over the random effect at the trial level and a Gaussian-Hermite quadrature to integrate over the random effects at the individual level. This integration method is less time-consuming compared to full Monte Carlo and full Gaussian-Hermite quadrature integration methods. However, users can change the method of integration in the event of convergence issues. This has already been discussed in the response to comment #1. As previously shown in the methodological paper, the method is quite robust to integration regarding the surrogacy evaluation criteria. Therefore, in the event of model convergence, we expect close results regarding the validation of surrogate endpoints.

STE:

6- What is IQWIG?

IQWIG stands for the German Institute for Quality and Efficiency in Health Care (Institut für Qualität und Wirtschaftlichkeit im Gesundheitswesen). It is an independent health technology assessment agency that assesses the benefits and harms of drug and non-drug technologies on behalf of the German Federal Joint Committee and the Federal Ministry of Health. It has issued recommendations for the evaluation of surrogate endpoints. 

For the sake of clarity, we have replaced “IQWIG” in the manuscript by “German Institute for Quality and Efficiency in Health Care (IQWiG)”

Computational details:

7- Not all researchers can afford a Xeon with 40 cores and 300+ Gb of RAM. Could you elaborate on the computational requirements of this method? Would it be possible to fit any model on e.g. a standard laptop or desktop PC?

We also tested the application on real data on a standard laptop and a desktop PC under recent versions of R and obtained exactly the same results, but with longer computing time. We have added the following to the section “Computational details and package installation”:

“A standard laptop and a desktop PC under recent versions of R can be used to fit the model. The results will be the same, but with an increase in computing time. For example, using a standard desktop PC in the application, the fit took around 1 hour compared to 9 min with a server including 40 cores and a RAM of 378 Go.”

8- You are using R 3.4.3, which was released almost 2 years ago. How does the software run with newer versions of R?

Now, we are using R 3.5.2 and the results are the same. We have updated the R version in the manuscript to “3.5.2” instead of 3.4.3

9- You mention that dependencies must be installed. Could you describe them, to make the readers aware?

We have rephrased package installation process as follows:

“The frailtypack package can be installed in any R session using the install.packages command as follows:

install.packages("frailtypack", dependencies = T, type ="source",repos = "https://cloud.r-project.org")

Installation via GitHub is possible thanks to the devtools package. All dependencies required by frailtypack must be installed first. The installation commands are:

install.packages(c("survC1","doBy","statmod"), repos = "https://cloud.r-project.org")

devtools::install_github("socale/frailtypack", ref = "surrogacy_submetted_3-0-3")

Finally, frailtypack must be loaded using the command:

library(frailtypack)”

Data:

10- Minor comment: data(“dataOvarian”) only works if frailtypack is loaded first; you could add the “package” argument to make the requirement explicit: data(“dataOvarian”, package = “frailtypack”)

You are right. We have rephrased the command as follows:

“data(“dataOvarian”, package = “frailtypack”)”

Surrogacy evaluation:

11- The arguments of “jointSurroPenal” that were set when fitting the joint frailty model with the ovarian cancer dataset are described but not motivated. Why were those specific values chosen? Does this affect the results of the joint model? If so, how? Model misspecification is a serious issue that can lead to biased estimates of the treatment effect, also in frailty models.

We provide a complete answer to this comment in the response to comments #1 and #2. In addition, we have updated the manuscript accordingly. We observed in simulation that model misspecification did not really affect the fix treatment effects (Sofeu et al., 2019). However, given the research question, we studied in depth the effect of misspecification on surrogacy evaluation criteria (Kendall’s Tau and R2trial).

Choice of the model based on LCV:

12- To me, the two models don’t give similar results. For instance, the estimated fixed treatments effects are quite different (e.g. -2 vs -1.6 for beta_S). This ties with some of my previous comments on model misspecification and the choice of the estimation arguments when fitting the joint frailty model.

We thank the reviewer for this remark and hope that the responses above have helped to dispel any doubts on model misspecification and the choice of the arguments/values for the jointSurroPenal() function. We observed in simulation that the model was quite robust asymptomatically, and also with estimation of fixed treatment effects. Based on a dataset, for two distinct models a slight difference can be observed between some points estimates. However, the 95% confidence intervals for these estimates overlapped. This requires the goodness of fit to be studied in order to choose the best model using the LCV criteria. 

Simulation studies:

13- I don’t understand the utility of this section. Could you elaborate a bit more on that? If you simulate data from a joint frailty model and then fit a joint frailty model with the same model formulation to the simulated data, then good performance is expected. I am probably missing the point here (sorry for that), but it would be good to describe in more detail why and when this is useful.

The aim of the simulation studies section is to illustrate the jointSurroPenalSimul() function. As describes in the discussion section, this function can help in planning a new trial and defining the optimal number of clusters when evaluating surrogate endpoints given the joint surrogate model. For example, if a given meta-analysis includes few trials, simulation studies may guide the choice of the minimum number of centers to be considered for a better estimation of the surrogacy evaluation criteria. Jurgen et al. (2015) suggested using clinical trial simulations to optimize adaptive trial designs. As they explained, the typical goal of clinical trial simulation is to identify a design that has a high probability of success based on the most likely conditions but which can also perform well, or at least acceptably, under more extreme conditions if necessary. In addition, if the recommended values for the arguments do not make it possible to reach convergence or involve a longer computer time when fitting the joint surrogate model, simulation studies can help given the data characteristics to choose optimal values for the number of quadrature nodes, the number of samples for the Monte-Carlo integration or for the number of nodes for splines. For the sake of clarity, we have reworded the paragraph about the usefulness of the jointSurroPenalSimul() function in the discussion as follows:

“Moreover, thanks to the jointSurroPenalSimul() function, it is possible to perform simulation studies in order to plan a new trial and define the optimal number of clusters when evaluating surrogate endpoints given the joint surrogate model. For example, if a given meta-analysis includes few trials, simulation studies may help in establishing the minimum number of centers to obtain the best estimate of the surrogacy evaluation criteria. Jurgen et al. (2015) suggested using clinical trial simulations to optimize adaptive trial designs. As they explained, the typical goal of a clinical trial simulation is to identify a design that has a high probability of success based on the most likely conditions but which can also perform well, or at least acceptably, under more extreme conditions if necessary. Simulation studies can help if the recommended values for the arguments do not make it possible to reach convergence or involve longer computer time when fitting the joint surrogate model. Given the data characteristics, they can help in choosing optimal values for some arguments (the number of quadrature nodes, the number of samples for the Monte-Carlo integration and the number of nodes for splines) and in anticipating their impact on estimating the model parameters.

Ref

Jurgen H, Song W, John K. Using simulation to optimize adaptive trial designs: applications in learning and confirmatory phase trials. Clinical Investigation. 2015;5(4):401–413. doi:10.4155/CLI.15.14.

Discussion:

14- The possibility of choosing estimation arguments that affect convergence of the algorithm should be discussed more, including comparisons (e.g. numerical integration methods), drawbacks, and problems that may arise.

These points are now discussed in the manuscript, as responses to comments #1, #2, #3 and #7.

15- A comparison with other established methods to assess surrogacy should be discussed as well, motivating the use of joint frailty models. Furthermore, robustness of the software introduced with this manuscript should be discussed.

We have responded to this comment by responding to comment number #3.

Appendices:

16- There is a lot of material in the appendices - is it all necessary? I believe some of it is already included in the paper introducing the methodology (Sofeu et al., 2019, in Statistics in Medicine), and maybe readers could be referred to that instead.

We thank the reviewer for this remark. In the appendices S1A, S1B and S1C, we just recall the formulation of the penalized marginal log-likelihood, Kendall’s tau and R2trial. Full information on this point can be found in the methodological paper. We believe that this short recall in the appendix is necessary to understand the output of the jointSurroPenal() function. The rest of the material in the appendix is new and concerns the derivation of the surrogate threshold effect (STE) and the help on the parameters of all functions defined in the manuscript.

17- Finally, the paper is at times hard to read, with some typos and several sentences that could be edited to improve clarity (remember that PLOS ONE does not copyedit accepted manuscripts). I spotted some typos, which are included below:

As a result of your comment, the manuscript has been copyedited by a professional scientific copyeditor. 

Title: “frailtypack” should start with a lowercase letter since it is the name of the package;

Keywords: “fraity” instead of “frailty”; “surrogte” instead of “surrogate”; “envent” instead of “event”

We have edited the keywords in the submission process

Abstract: The first sentence is hard to read; “quiet” instead of “quite” in the background and objective section (line 9);

We corrected “quite” and rephrased the first sentence as follows:

“The use of valid surrogate endpoints can accelerate the development of phase III trials.”

Line 89: replace “discuss” with “discussed”

Done

Line 131: replace “measurements” with “measure”

Done 

Line 134: replace “interpreting” with “interpretation”

Done 

Line 136: replace “equals” with “equal”

Done 

Line 225: replace “describe” with “described”

Done 

Line 237: GitHub should have capitals G and H (it’s the name of the company/service)

Done 

Line 241: did you mean Gb instead of Go?

Exactly 

Line 342: replace “respectievely” with “respectively”

Done 

Line 365: spell out “loocv”

We replaced loocv by “leave-one-out cross validation criteria (loocv)”

Line 447: replace “censorship” with “censoring”

Done 

Line 448: replace “which” with “where”

Done 

Reviewer #2: Sofeu et al. presented a manuscript that is an R implemented of their method published in “One-step validation method for surrogate endpoints using data from multiple randomized cancer clinical trials with failure-time endpoints. Statistics in Medicine. 2019;1–15”, with some enhanced functions for the validation of candidate surrogate endpoints. The manuscript presented some potentially useful alternative method for validate surrogate endpoint, based on the method published in the associated Statistics in Medicine paper. This package could potentially help researchers to choose more appropriate statistical method when validate surrogate endpoints. However, there is lack of demonstration of the stated advantages in comparison to other methods, and detailed discussion of the potential benefit of using the stated method is missing. Only general comparison was presented in the “Discussion” section without much depth or details. As such, the stated advantages were not well established in this manuscript. Overall, the manuscript mostly demonstrated the technical side of the package. Therefore, this reviewer encourages the author to consider enhances the manuscript so a researcher (instead of a package user) can benefit from reading the manuscript and decided how to realize its potential statistical advantages when used appropriately.

Major comments:

1. Numerous English language grammar errors in the manuscript is a real problem. Some of them listed later in the minor comments. Many sentences are nearly incomprehensible. An English proof reader is in need.

As a result of your comment, the manuscript has been copyedited by a professional scientific copyeditor. 

2. In the introduction section, the authors briefly discussed the pros and cons of existing methods and their method for surrogate endpoint validation. Yet, the manuscript (and package) focused on meta-analysis of such data. It might be obvious to the authors why such data is well suit and/or needed, but it should be make clear that a) if the method can be used for single study/center data, and b) the rational of using meta-analysis data in the demonstration of the manuscript. This is vital for the readers to understand the applicability and limitations of their package/methods.

We thank the reviewer for this comment. The need for meta-analysis data when validating surrogate endpoints has been discussed by several authors (Buyse and Molenberghs 1998; Burzkwoski et al. 2005; Buyse et al. 2015; Paoletti et al. 2016). It is due to some practical problems encountered when the validation approach is based on a single trial and the need to take heterogeneity between trials into account, for the purpose of prediction outside the scope of the trial. The one-step validation approach by Sofeu et al. (2019) is based on meta-analytic (or multicenter) data. For this reason, we have included the following paragraph in the introduction of the manuscript for the sake of clarity. 

“Prentice (1989) enumerated four criteria to be fulfilled by a putative surrogate endpoint. The fourth criterion, often called Prentice’s criterion, stipulates that a surrogate endpoint must capture the full treatment effect upon the true endpoint. The validation of Prentice’s criterion based on a clinical trial was quite difficult, mainly due to a lack of power and the difficulty to verify an assumption related to the relation between the treatment effects upon the true and the surrogate endpoints. Therefore, to verify this assumption and obtain a consistent sample size, Buyse et al (2000) like other authors suggested basing validation on the meta-analytic (or multicenter) data. An important point when dealing with meta-analytic data is to take heterogeneity between trials into account, for the purpose of prediction outside the scope of the trial. Thus, a validated surrogate endpoint from meta-analytic data can be used to predict the treatment effect upon the true endpoint in any trial”

3. In the associated R package, the Vignettes did not provide enough details for even the minimal example. On the other hand, this manuscript provided only marginally more information than a package Vignettes. Some statistical insights and details would make this manuscript more useful for a researcher instead of an R package user, for example, practical considerations in model estimation, interpretation of the results (like the summary presented in page 12-13), and any potential issues when choose different parameters. In addition, any practical advantages (if any) derived from these results using the method for the Ovarian Cancer data presented here would be very useful in demonstrate the benefit of this method over others. If not yet available, at least discuss potential gains by using this method instead of the others.

To provide added value to this manuscript compared to a package vignette, we now discuss the choice of the arguments/values for the main function and the management of the convergence issues by varying values of the arguments, and the robustness of the model (see responses to comments #1, #2, #5, #9, #11 and #12 of reviewer #1). We also emphasize the benefit of proposing a function for simulation studies (see response to comments 13 to reviewer #1). The manuscript has been reworded accordingly. The outputs of the jointSurroPenal() function have been well documented in the section “Summary of results” after the illustration on the advanced ovarian cancer dataset. We have reworded this description for the sake of clarity.

Regarding the package vignette, it just describes the available models within the package with the corresponding options. However, we provide different demos for basic examples. If published, this manuscript will be referenced in the package to facilitate its usage. 

Regarding the benefits of the method over the others, they were addressed in the methodological paper (Sofeu et al. 2019). Compared to the one-step copula approach, the main advantages of our model are in the reduction of convergences and numerical issues, the robustness to model misspecification, the flexibility of the model, the validation in one step and the estimation of R2trial without need for adjustment on the estimation errors. However, we have now reworded the second paragraph of the discussion section of the manuscript in order to underline the main advantage of the package compared to the others as follows: 

“By varying the values of the arguments in the jointSurroPenal function, convergence of the model is not always guaranteed. Therefore, it is important in the event of convergence issues to know how to play with the couples arguments/values as shown in the previous section. Thus, users can choose the method of integration, initial values, the number of nodes for splines and the smoothing parameters, the number of nodes to use for the Gauss-Hermite quadrature and the number of samples to consider for the Monte-Carlo integration when applicable, the random number generator, and other necessary arguments. It is also possible to set some parameters of the model in the event of identifiability issues. This underlines the flexibility of the frailtypack package in managing convergence issues. This flexibility is quite different from that obtained with the surrosurv package or macros SAS, for evaluating surrogate endpoints using the two-step Copula model or one-step Poisson model. Other advantages of our model compared to existing approaches are in the reduction of convergences and numerical issues, the robustness to model misspecification, the surrogacy evaluation based on a one-step approach and therefore the estimation of R2trial without need for adjustment on estimation errors. In addition, as underlined in the illustration section, the interpretation of Kendall’s tau is different from what is done with the two-step copula approach”

We also note in the discussion that the package can be used beyond the scope of evaluating surrogate endpoints to estimate the fixed treatments effects, taking competing risks and the heterogeneities in the data into account.

4. It is not clear what the results in the “Simulation studies” section can be used in the context of the Ovarian Cancer data analysis, other than a brief sentence in the Discussion section. The author may want to explain how such results are useful for the stated purpose (plan future trial) there.

Although simulation studies can be used to choose optimal values to assign to some arguments of the jointSurroPenal() function when fitting the model on the advanced ovarian cancer meta-analysis dataset, their use in the manuscript is not directly due to this dataset. Simulation studies can be performed prior to any meta-analytic dataset. A consistent response to this comment can be found in the response to comment #13 of reviewer #1. We have updated the manuscript accordingly. 

Minor comments:

1. In Abstract the sentence “We have especially the surrogate threshold effect…” need revision to make a valid English sentence. Also “Other tools concerned data generation, studies simulation and graphic representations…”.

We have reworded these sentences.

2. Many minor grammar issues in main text.

The manuscript has now been copyedited by a professional scientific copyeditor.

Reviewer #3: This paper addresses an interesting problem, one which is numerically challenging to solve. I have comments concerning the presentation, the statistical problem, and the software. The last of these three is separated out and should be viewed as comments on possible future evolution of the package, but is not relevant to any editorial decisions about the current manuscript, which deals with the present offering of the package.

1. I found the Introduction confusing. The title of the paper leads one to believe that this is a description of a particular package, it needs an early lead in to the fact that this is NOT what the paper is about. Two or three sentences would do, e.g., "frailtypack is an R package that fits a variety of survival models containing one or more random effects, or frailties...." "it includes for instance simple shared frailty, correlated...." "...for this paper we will focus on a particular subset of features applicable to the evaluation of surrogate endpoints ..." We need just enough to orient the user so that they don't feel like they were dropped into chapter 2 of a novel.

We thank reviewer #3 for this comment. For the sake of clarity, we have modified the title of the manuscript. The new one is “How to use frailtypack for validating failure-time surrogate endpoints using individual patient data from meta-analyses of randomized controlled trials”. Also, we have added the following paragraph in the introduction section of the manuscript in order to guide the user to the part of the package concerned:

“frailtypack is an R package that fits a variety of frailty models containing one or more random effects, or shared frailty. For instance, it includes a shared frailty model, a joint frailty model for recurrent events and terminal event, others forms of advanced joint frailty models (Krol et al. 2017), and now a joint frailty model for evaluating surrogate endpoints in meta-analyses of randomized controlled trials with failure-time endpoints. In this paper we focus on a particular subset of features applicable for evaluating surrogate endpoints.”

2. The description of the competing methods, starting at about page 2 line 14, was quite difficult to follow. The paragraphs need "roadmap" sentences to help readers who have not immersed themselves in the field know what the goal of the journey (paragraph) will be. It is hard to know what is background and what is essential. (This reviewer for instance -- I know some of the overarching questions but it is not a personal research area.) Without that the information is both too much and too little.

We now include a paragraph before the description of the competing methods to explain the problem with a single trial and the reason for the meta-analysis validation approach (see response to comment #2 of the reviewer #2). The manuscript has now been copyedited by a professional scientific copyeditor:.

(page 3, lines 78-85 are nicely done.)

Thank you.

3. Equation 1 is surprising -- where are the covariates? (See line 17, which introduced a need for them.) I realize that in practical work there will be at most 1 or 2 prognostic factors that are widely enough recognized such that all studies will have gathered them. Can you sidestep the issue by creating extra strata?

In Equation 1, we include one covariate: the treatment indicator (Z_ij1) with a fixed effect and a random effect treatment-by-trial interaction to deal with the heterogeneity at the trial level in interaction with the treatment. This is different from the trial-specific treatment effects considered in the first stage of the two-step copula model of Burzykowski et al. (2001). Another advantage of our approach compared to the two-step approach is to take the potential prognostic factors not observed in the dataset into account through the individual-level random effects. However, for the purpose of validating surrogate endpoints, it is not necessary to adjust the model on additional prognostic factors. However, we take your point and plan to include in the package the possibility to adjust on more prognostic factors.

When Z is a 0/1 covariate, as is often the case, then $\\v_S$ and $v_T$ are not identifiable for the control subjects. Does this cause issues wrt estimation of the variance of $v$?

We did not experience any problems of identifiability for estimating the variance covariance matrix (sigma_v) of the random effect treatment-by-trial interaction, given that a meta-analysis of randomized controlled trials as considered here always includes untreated and treated subjects. Therefore, there is always information for estimating sigma_v.

4. The function on lines 163 and following needs some explanation. First there is no formula. Only later when reading the examples, was this reviewer able to guess that you decided that the user should use a certain set of predetermined variable names. Please add some text.

We refer users to the Appendix for details on the jointSurroPenal() function. This function does not need formula given that formula is implicit when the dataset is named as we recommend. In the illustration section of the manuscript, we have added comments to guide users on the choice of the arguments for this function, when responding to comment #1 of the reviewer #1. Moreover, we have reworded the comment of this function as follows: 

“The mandatory argument of this function is data, the dataset to use for the estimations. Argument Data refers to a dataframe including at least 7 variables: patienID, trialID, timeS, statusS, timeT, status and trt. The description of these variables like other arguments of the function can be found in S2A Appendix in S2 Appendix, or via the R command help(jointSurroPenal). The rest of the arguments can be set to their default values. In addition, details on the required arguments/values are given in the illustration section” 

(Software suggestion: There list of options is terribly long; how is one to know what is necessary and what is optional. The pattern found in glm.control and many other packages would be a good way to separate out the secondary ones.)

We now include comments about the necessary and optional arguments in the illustration section (see response to comment #1 of the reviewer #1). 

5. The double hash signs on the left of the printout are a distraction and should be removed. (When there is lots of code, little output, and the goal is to allow users to easily cut/paste, then this peculiar design choice is defensible. None of those 3 is true here.)

Done

6. The discussion of the results on pages 10-11 is an important part of the paper. The authors have several good choices in the material just prior to this section wrt moving significant portions of it to the appendix. However, several details of the discussion are very terse. For example the statements at lines 298-302. Why is a value of 6.8 "strong" and a value of 1.79 "more pronounced".

We thank the reviewer for the comment. We now include more details for the sake of clarity. Therefore, the new comments should improve the understanding of this important part of the manuscript.

7. A first question of an estimation method is whether it works when the data set exactly fits the proposed model (if that fails then an approach is nearly useless). The far more important one is how the method works when the model is not quite true, i.e., the case for any real data set. The package's simulation modules are designed to address only the first of these, not the second, which is a serious limitation. More serious is the lack of any evaluation of the approach outside of the ideal case. Some methods are methods are robust to such changes and some are fragile. 

The robustness of the method to model misspecification had been studied in simulation studies and was detailed in the methodological paper published in Statistics in Medicine (Sofeu et al. 2019), and we found satisfactory results. The current manuscript aims at wider outreach for the R package that implements this model. However, we plan to include an option in the jointSurroPenalSimul() function for simulation studies in which survival times are generated from other distributions (e.g. Copula model, Poisson model). We now include some discussion in the manuscript regarding this comment, as a response to comment #2 of the reviewer #1.

8. Figure 1 is not very helpful, and is in fact confusing. Arrows normally go from parent to child. Do you really mean to imply that the plot() function generates data that is used by the estimation functions? Why is there an arrow out the side of summary()?

As we noticed in the legend of Figure 1, the direction of the arrow indicates that the object from the parent function is used by the child function. Therefore, the plot() function uses the object from the jointSurroPenal() function. However, you are right about the usefulness of this figure since all functions have already been defined. We have therefore removed it from the manuscript.

Figures 2 and 3 are at odds. The paper states that they are two "representations" of a fit to the same data, yet Figure 2 has median values of 2.5 and 4 months, while figure 3 shows values close to .5 and 1 year. If these are simply two different solution paths, it would argue that the software is very unstable. No discussion is provided to guide the user with respect to these discrepant results.

As indicated in the legends, Fig 2 is based on the advanced ovarian cancer meta-analysis, while Fig 3 is based on the generated data. In addition, the following comment had already been included in the manuscript in order to avoid confusion:

“Fig 3 shows another representation of the baseline survival and hazard functions for the surrogate and the true endpoints. We use the object joint.surro.sim.MCPGH for this purpose” 

We have added the expression “which is based on the generated data” at the end of the previous comments.

Minor comments (no response required)

Abstract sentence 1: " ... for accelerated effectively the phase 3 trial." I've read this 3 times and still do not know what the words mean.

The sentence has been reworded as follows:

“The use of valid surrogate endpoints can accelerate the development of phase III trials”

Abstract: "This model was quiet robust..." I think you mean 'quite robust'.

Exactly 

$\\alpha$ and $\\zeta$ might make a little more sense attached to S rather than T. After all, death is death, but different

---

## [Decision Letter · Decision Letter 1]

12 Dec 2019

PONE-D-19-20769R1

How to use *frailtypack* for validating failure-time surrogate endpoints using individual patient data from meta-analyses of randomized controlled trials

PLOS ONE

Dear Mr. SOFEU,

Thank you for submitting your manuscript to PLOS ONE. After careful consideration, we feel that it has merit but does not fully meet PLOS ONE’s publication criteria as it currently stands. Therefore, we invite you to submit a revised version of the manuscript that addresses the points raised during the review process.

We would appreciate receiving your revised manuscript by Jan 26 2020 11:59PM. To enhance the reproducibility of your results, we recommend that if applicable you deposit your laboratory protocols in protocols.io, where a protocol can be assigned its own identifier (DOI) such that it can be cited independently in the future. For instructions see: http://journals.plos.org/plosone/s/submission-guidelines#loc-laboratory-protocols

We look forward to receiving your revised manuscript.

Kind regards,

Joshua Jones

Academic Editor

PLOS ONE

Journal Requirements:

Additional Editor Comments (if provided):

Please attend to the minor comments of the reviewers.

Reviewers' comments:

Reviewer's Responses to Questions

**Comments to the Author**

1. If the authors have adequately addressed your comments raised in a previous round of review and you feel that this manuscript is now acceptable for publication, you may indicate that here to bypass the “Comments to the Author” section, enter your conflict of interest statement in the “Confidential to Editor” section, and submit your "Accept" recommendation.

Reviewer #1: (No Response)

Reviewer #2: All comments have been addressed

2. Is the manuscript technically sound, and do the data support the conclusions?

Reviewer #1: Yes

Reviewer #2: Yes

3. Has the statistical analysis been performed appropriately and rigorously? 

Reviewer #1: Yes

Reviewer #2: Yes

4. Have the authors made all data underlying the findings in their manuscript fully available?

Reviewer #1: Yes

Reviewer #2: Yes

5. Is the manuscript presented in an intelligible fashion and written in standard English?

Reviewer #1: Yes

Reviewer #2: Yes

6. Review Comments to the Author

Reviewer #1: I would like to thank the authors for their thorough responses to reviewers’ comments - I think the manuscript has greatly improved, and my previous comments have been mostly addressed. Well done.

However, I still have some (minor) comments:

IQWiG is first introduced in line 111, but the acronym is introduced only in line 199 - I would move the acronym to line 111 instead.

There is some style inconsistency: some R output is coloured, some is not. Please make consistent throughout the manuscript, I am not sure if this will be copy-edited later on.

Line 287: it is a bit confusing what figure are you referring to. Please reword the sentence.

In computational details and package installation: it is not necessary to specify the repos argument when installing a package. Furthermore, the {devtools} functionality to install packages from GitHub seems to have been moved to the {remotes} package (and only imported in {devtools}). I would consider referring to {remotes} rather than {devtools}, but this is completely up to you.

I just noticed that the column with the patient ID is assumed to be “patienID”. Should it be “patientID” instead? This would be more coherent with the other names, where the whole word is used.

In the section on choosing model via the LCV: I think it is still a bit difficult to compare the estimates of the two competing models. I think it might be helpful to include a separate table that compares them side by side. Besides that, I still think that the coefficients are not so similar - including the confidence intervals that show large overlap would be great too, and would sell the point better (in my opinion).

Some sentences are a bit confusing at the moment, please reword:

Lines 365-366;

Line 382: “observation on...” could be replaced by “the estimated value of…”;

Line 389: reword after the comma;

Lines 508-510.

Finally, I found some more typos:

Line 103, replace “random effect” with “random effects”

Lines 380-381, replace “compare” with “compared”

Line 552, replace “neccesary” with “necessary”

Reviewer #2: Much more details had provided in the revision, and the involvement of professional typewriter improved the readability noticeably. The revision addressed all my concerns.

7. PLOS authors have the option to publish the peer review history of their article (what does this mean?). If published, this will include your full peer review and any attached files.

Reviewer #1: No

Reviewer #2: No

---

## [Author Response · Author response to Decision Letter 1]

6 Jan 2020

PONE-D-19-20769R1

How to use frailtypack for validating failure-time surrogate endpoints using individual patient data from meta-analyses of randomized controlled trials

We thank the editor and the reviewers for the great interest to our manuscript. Please find below our replies to the reviewers’ comments. We have also revised the manuscript in accordance with the comments when necessary.

6. Review Comments to the Author

Reviewer #1: I would like to thank the authors for their thorough responses to reviewers’ comments - I think the manuscript has greatly improved, and my previous comments have been mostly addressed. Well done.

However, I still have some (minor) comments:

IQWiG is first introduced in line 111, but the acronym is introduced only in line 199 - I would move the acronym to line 111 instead.

Done

There is some style inconsistency: some R output is coloured, some is not. Please make consistent throughout the manuscript, I am not sure if this will be copy-edited later on.

Thank you to reviewer #1 for this comment. Actually, only the R codes used for the outputs are colored. The uncolored codes are related to the definition (or description) of the corresponding functions. This is the case for all functions in section “Available functions in the frailtypack R package for surrogacy evaluation” and the S3method plot() in section “Illustration”. We chose not to color the outputs.

Line 287: it is a bit confusing what figure are you referring to. Please reword the sentence.

We reworded the sentence as:

“A list of other models implemented in frailtypack [23] can be found in S1 Fig”

In computational details and package installation: it is not necessary to specify the repos argument when installing a package. Furthermore, the {devtools} functionality to install packages from GitHub seems to have been moved to the {remotes} package (and only imported in {devtools}). I would consider referring to {remotes} rather than {devtools}, but this is completely up to you.

You are right concerning the points mentioned above. However, the repos argument helps to guide the user on the repository that we are currently used to install the package and to avoid on some platforms (such as in a Linux terminal) the selection in a proposed list the repository to use. Moreover, the use of this argument does not negatively impact the functionality of the command. Regarding the use of the package “remotes” rather than “devtools”, we have not experienced the first one and as mentioned by the reviewer, this package is imported in “devtools”. So, we chose to use the “devtools” package since it allows several tools necessary for the compilation of the package.

I just noticed that the column with the patient ID is assumed to be “patienID”. Should it be “patientID” instead? This would be more coherent with the other names, where the whole word is used.

We rewrote the column name patienID of argument data as patientID. We updated the package and the manuscript subsequently

In the section on choosing model via the LCV: I think it is still a bit difficult to compare the estimates of the two competing models. I think it might be helpful to include a separate table that compares them side by side. Besides that, I still think that the coefficients are not so similar - including the confidence intervals that show large overlap would be great too, and would sell the point better (in my opinion).

We thank reviewer #1 for this comment. We aim in this section to illustrate how to use LCV for model comparison. It is not really the goal to compare the estimates of the two competing models, given that they can be different. So by using LCV, one can choose the best model according to the data. We rephrased the interpretation in the corresponding section as follows:

“As expected [17], the two observed values of LCV are quite similar. The summary() function applied to previous objects give results shown below. When comparing the two models, estimates of most coefficients and standard errors showed some differences. However this observation does not alter conclusions on the surrogacy validity captured by Kendall's tau and R2trial.”

Some sentences are a bit confusing at the moment, please reword:

Lines 365-366;

We rephrased the sentence as:

“By changing the value of some arguments, we obtained the following set of arguments/values which allowed convergence:”

Line 382: “observation on...” could be replaced by “the estimated value of…”;

Done

Line 389: reword after the comma;

We rephrased the sentence as:

“These parameters can be interpreted as usual, but taking adjustment on the random effects into account.”

Lines 508-510.

We rephrased the sentence as:

 “By varying the values of the arguments in the jointSurroPenal function, convergence of the model is not always guaranteed. Therefore, it is important in the event of convergence issues to know how to play with the arguments/values couple as shown in the previous section.”

Finally, I found some more typos:

Line 103, replace “random effect” with “random effects”

Done 

Lines 380-381, replace “compare” with “compared”

Done 

Line 552, replace “neccesary” with “necessary”

Done 

Reviewer #2: Much more details had provided in the revision, and the involvement of professional typewriter improved the readability noticeably. The revision addressed all my concerns.

Thank you

---

## [Editor Report · Decision Letter 2]

8 Jan 2020

How to use *frailtypack* for validating failure-time surrogate endpoints using individual patient data from meta-analyses of randomized controlled trials

PONE-D-19-20769R2

Dear Dr. SOFEU,

We are pleased to inform you that your manuscript has been judged scientifically suitable for publication and will be formally accepted for publication once it complies with all outstanding technical requirements.

With kind regards,

Alan D Hutson

Academic Editor

PLOS ONE
---

## [Editor Report · Acceptance letter]

13 Jan 2020

PONE-D-19-20769R2 

How to use frailtypack for validating failure-time surrogate endpoints using individual patient data from meta-analyses of randomized controlled trials 

Dear Dr. Sofeu:

I am pleased to inform you that your manuscript has been deemed suitable for publication in PLOS ONE. Congratulations! Your manuscript is now with our production department. 

With kind regards,

on behalf of

Dr. Alan D Hutson 

Academic Editor

PLOS ONE